# Thinking as Society: Multi-Social-Agent Self-Distillation for Multimodal Misinformation Detection

**Yifei Gao, Ning Xu,**\* **Wenhui Li, Hongshuo Tian, Lanjun Wang & An-An Liu**\*
Tianjin University
Tianjin, China
`ningxu@tju.edu.cn, anan0422@gmail.com`

## Abstract

Multimodal Misinformation Detection (MMD) in realistic, mixed-sourced scenarios must incorporate robust reasoning capabilities to handle the social complexity and diverse types of forgeries. While MLLM-based agents are increasingly used for MMD task due to their powerful reasoning abilities, they suffer from a critical trade-off: on one hand, single-agent methods provide only the limited, single-view analysis; on the other hand, multi-agent methods introduce high computational costs and significant optimization difficulties. To address this gap, we propose a novel Multi-Social-Agent Self-Distillation framework that internalizes collective social reasoning capabilities into a unified model. Our framework consists of two core stages: (1) we simulate multi-perspective judgments from a diverse society of MLLM agents and synthesize their collective feedback into high-quality Social Chain-of-Thought (SCoT) data; (2) Building on this, we propose the Social Correction Value-Driven Preference Optimization (SCPO), a new alignment algorithm that leverages the degree of social misjudgment as a verifiable signal to dynamically focus training on the most challenging samples. Extensive experiments on the challenging MFC-Bench and MMFakeBench benchmarks demonstrate the effectiveness of our framework. Our 7B Qwen2-VL-based model significantly outperforms various MLLM baselines, multi-agent methods, and even competes or surpasses proprietary models like GPT-4o and Claude, facilitating advanced multimodal misinformation reasoning and detection via thinking as society.

## 1 Introduction

Multimodal misinformation, including fake news, often contains convincing yet deceptive content that spreads easily across social networks Vosoughi et al. (2018). In particular, recent generative models like GPT OpenAI (2024) and Stable Diffusion Rombach et al. (2022) have made it easier to create such misleading multimodal content Chen & Shu (2024). This realistic, mixed-sourced misinformation often combines diverse forgery types and leverages social context, posing a serious threat to public trust Akhtar et al. (2023). Therefore, effectively addressing these threats requires detection models that go beyond just basic classification Qi et al. (2024). They must be capable of understanding social dynamics, performing robust reasoning and making comprehensive judgments Wang et al. (2025) to handle such complexity and achieve generalization, robustness and interpretability.

Recently, MLLM-based agents are increasingly adopted for Multimodal Misinformation Detection (MMD) owing to their powerful reasoning and generalization capabilities Liu et al. (2025b); Cekinel et al. (2025); Tahmasebi et al. (2024); Lee et al. (2024). However, as shown in Fig.1, they suffer from a critical trade-off in their application: on one hand, single-agent methods Cekinel et al. (2025); Tahmasebi et al. (2024) provide only a limited, single-view analysis, which is insufficient for socially complex tasks and easy to be misled. On the other hand, while multi-agent methods Liu et al. (2025b); Wan et al. (2024); Nan et al. (2024) can conduct analysis from diverse social roles, they consume extensive computational resources and have difficulty for end-to-end optimization. This

---

\*Corresponding authors.

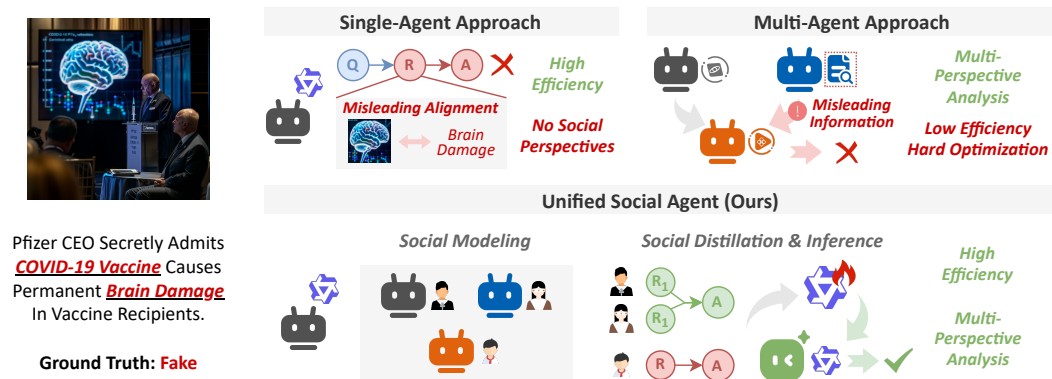

Figure 1: Overview of existing MLLM-based detection paradigms. The single-agent approach (top left) is efficient but lacks diverse social perspectives, which can be easily misled. The multi-agent approach (top right) captures multiple viewpoints but suffers from low efficiency and optimization challenges. In contrast, our method leverages social modeling and distillation to build a unified social agent that combines the high efficiency with the robust reasoning capability.

dilemma between inference efficiency and multi-perspective analysis motivates the requirement for a new framework that can internalize collective social reasoning within a unified model.

This objective to internalize the collective reasoning exposes two fundamental challenges. **(1) Data Limitation:** the lack of rich and multi-perspective reasoning data to teach MLLMs how to integrate diverse social viewpoints and apply valuable evidence to improve decision-making; **(2) Optimization Challenge:** existing fine-tuning algorithms like SFT and DPO Rafailov et al. (2023) fail to distinguish and prioritize high-value samples reflecting social cognitive differences. Consequently, even after building high-quality reasoning data, training results remain suboptimal.

In response to these challenges, we introduce a novel **Multi-Social-Agent Self-Distillation** framework. For the *data limitation*, we simulate a diverse set of MLLM agents, each providing its own perspective to assess the authenticity of given multimodal content. Then, we integrate all of their feedback to create high-quality Social Chain-of-Thought (SCoT) data. For the *optimization challenge*, we propose a new fine-tuning algorithm, **Social Correction Value-Driven Preference Optimization (SCPO)**. Building on our SCoT data, SCPO quantifies the degree of social misjudgment into an explicit and verifiable value. This value helps the model dynamically adjust optimization process to focus on the most challenging samples with cognitive gaps. By incorporating this social correction value into preference optimization process, our approach achieves an effective and stable training procedure. Through our multi-social-agent self-distillation framework, we can achieve "thinking as society", enabling a single model to reason from the perspectives of multiple social roles, thereby realizing deep understanding and decision-making for the MMD task.

We conduct a comprehensive evaluation on two latest MLLM-oriented MMD benchmarks: MFC-Bench Wang et al. (2025) and MMFakeBench Liu et al. (2025b). Our 7B Qwen2-VL-based model significantly outperforms stronger open-source MLLM baselines Wang et al. (2024a); Bai et al. (2025); Zhu et al. (2025), multi-agent frameworks Liu et al. (2025b), and even competes or surpasses proprietary models like GPT-4o and Claude, facilitating advanced reasoning and detection for multimodal misinformation in challenging mixed-sourced scenarios via thinking as society.

In summary, our main contributions are: (1) We introduce a novel multi-social-agent self-distillation framework for advanced reasoning and detection of realistic multimodal misinformation, resolving the critical trade-off between the single-agent limited perspective and multi-agent inefficiency. (2) We develop a complete self-distillation pipeline to generate high-quality Social Chain-of-Thought (SCoT) data by simulating and synthesizing multi-perspective social feedback, addressing the data limitation. (3) We propose the Social Correction Value-Driven Optimization (SCPO), a new alignment algorithm that introduces social misjudgement as a verifiable signal to prioritize challenging training samples, solving the optimization challenge. (4) Extensive experiments on MFC-Bench and MMFakeBench demonstrates our 7B Qwen2-VL-based model significantly surpasses larger open-source models, dedicated multi-agent frameworks and powerful proprietary systems.

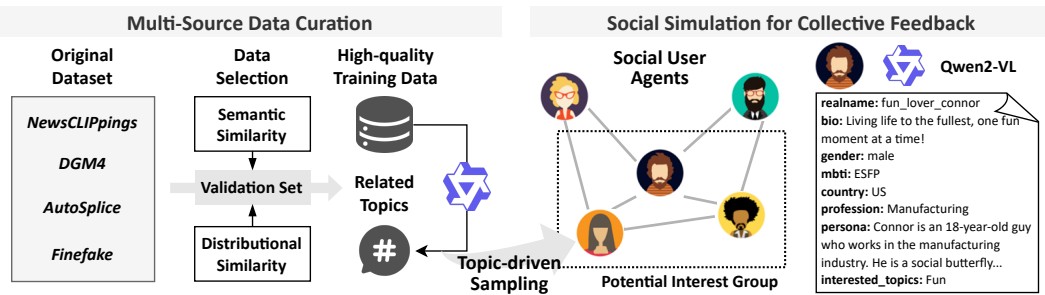

Figure 2: Overall pipeline of multi-source data curation and social simulation.

## 2 RELATED WORKS

**Multimodal Misinformation Detection.** Multimodal misinformation detection is to automatically identify false or misleading contents of different modalities Abdali et al. (2025). Early researches focus on developing specialized multimodal feature extraction Abdelnabi et al. (2022); Khattar et al. (2019); Wang et al. (2018), integrating other pattern analysis such as writing style Przybyla (2020) and sentiments Ghanem et al. (2021). In addition to content-based analysis, due to the social attributes of misinformation such as news domains Luo et al. (2021); Shao et al. (2023); Zhou et al. (2024), some works explore social-context-based detection, such as propagation patterns Zhou & Zafarani (2019); Xu et al. (2024), user feedback Min et al. (2022), and social networks Nguyen et al. (2020). Recently, some works apply MLLMs to knowledge bases Cekinel et al. (2025); Tahmasebi et al. (2024), simulated comments Wan et al. (2024); Nan et al. (2024), and rationale generation Lee et al. (2024); Hu et al. (2024), enhancing detection accuracy and explanability. The wide application of MLLMs also leads to more comprehensive benchmarks like MFC-Bench Wang et al. (2025) and MMFakeBench Liu et al. (2025b), designed to evaluate the reasoning and detection capabilities required by mixed-source multimodal misinformation. Our work enhances the intrinsic reasoning of MLLMs for complex real-world multimodal misinformation detection.

**Preference Optimization for MLLM Alignment.** Aligning MLLMs with human preferences is critical for ensuring reliable deployment Shi et al. (2024); Sun et al. (2024). Traditional Reinforcement Learning from Human Feedback (RLHF) includes reward model training and reinforcement learning Ouyang et al. (2022). To address the training instability and computational costs in RLHF, Direct Preference Optimization (DPO) Rafailov et al. (2023) reframes alignment as supervised learning without explicit reward model. This innovation has inspired numerous variants such as TDPO Zeng et al. (2024b), SimPO Meng et al. (2024), ORPO Hong et al. (2024) and TIS-DPO Liu et al. (2025a). DPO framework also has wide application in multimodal domain. For example, MM-DPO Zhang et al. (2025) uses reward margins to control the preference strength. Our SCPO introduces a verifiable and socially-grounded signal to adjust training priority dynamically.

## 3 METHODOLOGY

In this section, we detail our multi-social-agent self-distillation framework which consists of two parts, i.e., the pipeline of multi-agent social chain-of-thought generation (Sec 3.1) and the algorithm of social correction value-driven preference optimization (Sec 3.2).

### 3.1 MULTI-AGENT SOCIAL CHAIN-OF-THOUGHT GENERATION

#### 3.1.1 MULTI-SOURCE DATA CURATION AND SOCIAL SIMULATION

To generate high-quality SCoT data, our framework first requires two foundational components: a diverse set of multimodal misinformation to employ social discussion, and user agents with related interests and various backgrounds to provide realistic social feedback.

**Misinformation Curation and User Profiles.** To ensure the diversity and representativeness of multimodal misinformation, we aggregate multiple existing MMD benchmarks into a comprehen-

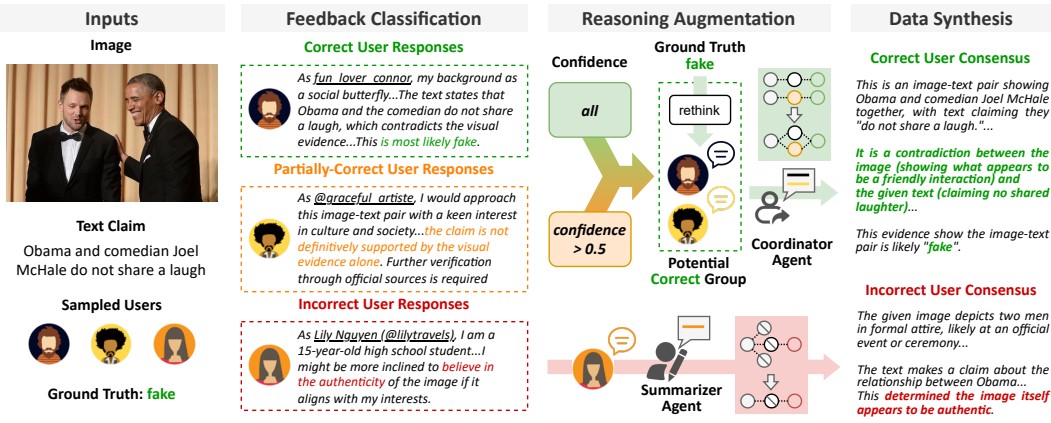

Figure 3: Overall pipeline of social chain-of-thought generation.

sive dataset. These include both synthetically generated examples via controlled manipulation (e.g., NewsCLIPpings Luo et al. (2021), DGM4 Shao et al. (2023), AutoSplice Jia et al. (2023)) and real-world samples collected from social media (e.g., FineFake Zhou et al. (2024)). To handle scale and redundancy, we rank the training samples based on metrics relative to their validation set, including both CLIP semantic similarity Radford et al. (2021) and Wasserstein gradient-based distributional consistency Kolouri et al. (2017). By top selection and label balancing, we obtained a high-quality, diverse misinformation base-dataset.

To capture social judgment diversity, we collect a large set of user profiles for realistic simulation. As shown in Fig.2, user agents are constructed based on the profiles with a wide range of demographic (e.g., global representation across countries), psychological (e.g., enthusiastic, calm), professions (e.g., journalists, engineers, educators), and interest-based attributes (e.g., politics, science) from OASIS platform Yang et al. (2024). This breadth allows for the feedback with distinct social roles and cognitive backgrounds, facilitating the reasoning of collective intelligence.

**Topic-Driven User Sampling.** To simulate contextually relevant social interactions, it is crucial to model how users engage with specific content. On social platforms, users preferentially interact with content aligned with their personal interests. To mirror this dynamic, we sample users who offer meaningful perspectives through topic-based matching.

The sampling process involves three steps. First, for each multimodal sample in the base-dataset, we extract its topic categories (e.g., a news article about "AI deepfakes in political campaigns" might be tagged with both "Politics" and "Information Technology"), denoted as $T_s$. Next, for each user with interested topic set $T_u$, we compute a matching score $w$ using the normalized overlap between $T_u$ and $T_s$: $w = \frac{|T_u \cap T_s|}{|T_s|} + \epsilon$, where $\epsilon$ is a small positive constant ensuring a non-zero selection probability for all users. This simulates the real-world scenario where users occasionally engage with content outside their primary interests. Finally, we perform weighted sampling from the user pool based on matching scores. This approach ensures users with higher topic relevance are more likely to be selected, while still introducing a controlled randomness of social media interactions.

For each sampled user, we instantiate an MLLM Wang et al. (2024a) agent to simulate their social role. This is achieved by injecting the user's profile into a structured system prompt (See specific prompts in A.7), which guides the agent to generate feedback consistent with its background and perspectives. Each feedback reflects a unique reasoning pattern, capturing the agent's potential biases and background-influenced judgments. This collection of multi-perspective feedback comprises the raw social reasoning data for our subsequent synthesis and optimization pipeline.

### 3.1.2 Social Chain-of-Thought Synthesis

To convert the diverse but unstructured social feedback into high-quality reasoning data, we develop a multi-stage data generation pipeline for self-distillation. Instead of generating a single correct answer, we first use LLM-as-a-Judge paradigm to classify the full spectrum of social responses,

then deepen the thoughts of promising responses for quality and relevance, and finally synthesize these refined views into well-organized preference SCoT data for our optimization algorithm.

**Answer-Centric Feedback Classification.**   Rather than simple binary classification, our approach is designed to discover any potentially valuable insights from diverse social feedback. Specifically, we introduce the soft boundary through fine-grained verification principles and prompt LLM-judger to provide self-critiqued confidence scores. First, the LLM judger categorizes each user feedback into one of three distinct sets based on the consistency between the user conclusion and ground-truth answer. 1) *Correct set*: The conclusion is correct with supporting reasoning. 2) *Incorrect set*: The conclusion misaligns with the answer and the reasoning contains critical flaws (e.g., neglecting key multimodal evidence). 3) *Partially-correct set*: The feedback shows partial alignment, such as incorrect conclusions while containing valid observations. Second, the LLM-judger assigns a confidence score (ranging from 0 to 1) to each classification as judgment reliability. Finally, to form a solid basis for subsequent augmentation, we define the *potential correct group* as the subset of feedback from the *correct set* and *partially-correct set* with confidence over 0.5. This threshold effectively balances the inclusion of diverse perspectives with the filtering of noisy feedback.

**Role-Specific Reasoning Augmentation.**   To deepen and optimize the reasoning in the *potential correct group*, we employ a two-fold strategy that augments feedback with external knowledge and adaptively handles edge cases. First, we perform knowledge injection by providing the ground-truth answer to each user in the *potential correct group* and prompting them to refine their explanation from their assigned role's perspective. This process encourages more elaborate reasoning and the incorporation of previously overlooked evidence, while still preserving their unique role-specific insights. Second, we apply adaptive enhancement to edge cases to improve robustness. For *easy samples* (where all agents respond correctly), we generate adversarial negatives by prompting all agents to justify the opposite conclusion, a counterfactual process that exposes potential cognitive biases and prevents overfitting. For *hard samples* (where no agent respond correctly), we broadcast the answer and prompt all agents to construct valid role-specific reasoning toward the correct conclusion for the most difficult examples.

**Preference Data Synthesis.**   We synthesize the augmented feedback into the preference SCoT dataset by two specialized MLLM-driven agents. The **Coordinator Agent** is responsible for generating the positive (chosen) samples. It processes refined feedback from the *potential correct group* to generate a unified positive SCoT. By merging overlapping reasoning paths while preserving role-specific insights that uniquely contribute to final judgment, it yields fine-grained social reasoning chains as high-quality positive data. The **Summarizer Agent** constructs the negative (rejected) SCoTs. It processes responses from the *incorrect* set, and its goal is to capture the most representative mistakes for negative data by maintaining core misleading reasoning links and critical flaws[1].

Overall, the synthesized chosen and rejected SCoTs form the core preference dataset for our subsequent optimization, enabling MLLMs to reason complex tasks from diverse social perspectives.

### 3.2   SCPO: Social Correction Value-Driven Preference Optimization

Existing preference alignment frameworks like DPO Rafailov et al. (2023) typically treat all training samples with the same importance, making insufficient use of social cognitive differences. To effectively utilize the social attributes of our SCoT data for guiding MLLM optimization, we propose a novel learning framework, Social Correction Value-Driven Preference Optimization (SCPO). SCPO measures the degree of social misjudgement as a verifiable correction value to adjust the preference optimization dynamically, enhancing the efficiency and robustness of alignment process.

**Social Correction Value Function.**   The core mechanism in SCPO is to quantify the correcting requirements as sample's learning value based on social feedback. For a given multimodal sample $x$, we introduce the social correction value function $sc(x)$, which measures the sample difficulty as its priority for preference learning.

$$sc(x) = 1 - \left( \frac{N_C}{N} + \frac{N_P}{N - N_C} \cdot \frac{1}{N} \right)$$

---

[1]The detailed architecture of coordinator and summarizer agents are in Appendix A.3.1.

where $N$ denotes the total user numbers for the sample $x$, $N_C$ is the user numbers of *correct set*, and $N_P$ is the user numbers of *partially correct set*. The intuition behind $sc(x)$ is two-fold: The coarse-grained term $\frac{N_C}{N}$ captures the proportion of users who correctly judge the sample. Its higher value indicates easier samples with lower correction requirements. The fine-grained term $\frac{N_P}{N-N_C}$ accounts for partial correctness among the remaining users. It represents a discounted weight over $\frac{1}{N}$ to avoid overshadowing the dominant effect of correct users.

Overall, $sc(x)$ is a natural preference scaling signal. Specifically, samples with higher $sc(x)$ are more challenging with more incorrect responses. These samples are required to amplify preference strengths, making MLLMs focus on resolving social misjudgement. Otherwise, easier samples with widespread correct conclusion are downweighted, preventing overfitting to trivial cases.

**SCPO Loss Function.**   We design the SCPO loss function under the ORPO integration framework Hong et al. (2024) of Supervised Fine-Tuning (SFT) and Preference Optimization (PO). The SCPO loss is as follows:

$$\mathcal{L}_{\text{SCPO}} = \mathcal{L}_{\text{SFT}} + \lambda \cdot (1 + \omega \cdot sc(x)) \cdot \mathcal{L}_{\text{OR}}$$

$$\mathcal{L}_{\text{SFT}} = -\mathbb{E}_{(x,y_w)\sim\mathcal{D}_{\text{SFT}}} \left[ \log \pi_\theta(y_w \mid x) \right]$$

$$\mathcal{L}_{\text{OR}} = -\mathbb{E}_{(x,y_w,y_l)\sim\mathcal{D}_{\text{pref}}} \left[ \log \sigma \left( \log \frac{\pi_\theta(y_w \mid x)}{1 - \pi_\theta(y_w \mid x)} - \log \frac{\pi_\theta(y_l \mid x)}{1 - \pi_\theta(y_l \mid x)} \right) \right]$$

SFT loss $\mathcal{L}_{\text{SFT}}$ guides MLLMs to learn the social reasoning patterns in $\mathcal{D}_{\text{SFT}}$, which denotes the dataset of positive social CoTs synthesized by the coordinator agent. To further align MLLMs to avoid misleading reasoning traces, we introduce Odds Ratio Alignment Loss $\mathcal{L}_{\text{OR}}$, which compares the generation likelihood of high-quality ($y_w$) with flawed reasoning ($y_l$). $\mathcal{D}_{\text{pref}}$ is a dataset of preference pairs ($y_w, y_l$), where $y_w$ is a high-quality social CoT (from the *potential correct group*) and $y_l$ is a flawed reference given by summarizer agent (from the *incorrect set*).

**Social Correction-Driven Preference Scaling.** A key innovation of SCPO is its dynamic weighting of OR loss via the social correction value $(1 + \omega \cdot sc(x))$. Existing efforts have explored using reward margins to adjust sample prioritization under DPO framework Zhang et al. (2025). However, these methods rely on a reward model to learn implicit preference relationships, making the resulting margins unverifiable Lambert et al. (2025). In contrast, our social correction value function $sc(x)$ serves as a more reliable indicator for dynamic preference weighting. First, $sc(x)$ is computed from observable statistics of user feedbacks with clear semantics, making it verifiable and interpretable. Second, the bounded range of $sc(x) \in [0, 1]$ prevents overly aggressive training, ensuring optimization stability. Our SCPO makes MLLMs self-distill collective cognition effectively, enabling to reason complex multimodal tasks from diverse social perspectives.

## 4 EXPERIMENT

### 4.1 EXPERIMENTAL SETUP

**Benchmarks.**   Our multi-social-agent self-distillation framework is evaluated on two MMD benchmarks. **MFC-Bench** Wang et al. (2025) is a comprehensive benchmark for assessing the factual accuracy of MLLMs in multimodal fact-checking. It encompasses three key stages of verdict prediction: Manipulation Classification, Out-of-Context (OOC) Classification, and Veracity Classification, covering *35K* multimodal samples from diverse social backgrounds like world news. **MM-FakeBench** Liu et al. (2025b) focuses on mixed-source multimodal misinformation detection. It covers three critical misinformation sources: distortion of textual veracity, visual veracity and cross-modal consistency, containing 12 sub-categories under realistic mixed-source scenarios.

**Baselines.**   We uniformly use Qwen2-VL-7B-Instruct Wang et al. (2024a) in our self-distillation framework, and the baselines cover various MLLMs and multi-agent methods for comprehensive comparison. We evaluate main proprietary MLLMs like GPT-4o, Claude3.5-Sonnet and open-source MLLMs Yao et al. (2024); Wang et al. (2024b) including Qwen-VL Bai et al. (2023); Wang et al. (2024a); Bai et al. (2025), InternVL Chen et al. (2024); Zhu et al. (2025), LLaVA Liu et al. (2023; 2024); Li et al. (2025) series. Notably, we include MLLMs stronger than base Qwen2-VL, such as

Table 1: Comparison between our Qwen2-VL-based self-distillation framework and other MLLM baselines on MFC-Bench in open-prompting setting. The accuracy and macro-averaged F1 score(%) are reported as the metrics. The best and second test results are in bold and underlined, respectively.

| Models | Size | Manipulation | | OOC | | Veracity | | Overall | |
|---|---|---|---|---|---|---|---|---|---|
| | | Accuracy | F1 | Accuracy | F1 | Accuracy | F1 | Accuracy | F1 |
| *Proprietary Models* | | | | | | | | | |
| GPT-4o | - | **68.03** | **67.00** | **82.15** | **82.11** | 69.85 | 69.04 | **69.11** | **68.49** |
| Claude3.5-Sonnet | - | 65.39 | 61.51 | 75.80 | 75.34 | **80.68** | **78.26** | 66.85 | 64.32 |
| *Open-Source Models* | | | | | | | | | |
| InternVL | 25.5B | 53.77 | 53.38 | 63.15 | 62.28 | 66.55 | 65.65 | 55.03 | 54.35 |
| CogVLM | 17B | 58.85 | 54.30 | 53.85 | 47.30 | 57.30 | 60.51 | 58.47 | 54.44 |
| LLaVA-NeXT | 13B | 59.96 | 58.03 | 55.60 | 52.81 | 43.40 | 47.45 | 58.77 | 56.84 |
| Pixtral | 12B | 56.97 | 56.94 | 68.20 | 68.17 | 74.85 | 72.09 | 58.62 | 58.64 |
| MiniCPM-V-2.6 | 8B | 56.20 | 56.25 | 61.65 | 61.49 | 64.90 | 64.52 | 57.00 | 57.01 |
| LLaVA-OneVision | 7B | 59.38 | 57.04 | 58.25 | 55.01 | 45.25 | 49.40 | 58.51 | 56.27 |
| Qwen2-VL | 7B | 57.36 | 57.02 | 63.80 | 63.79 | 48.95 | 52.78 | 57.24 | 56.91 |
| Intern-VL2 | 8B | 59.21 | 57.82 | 62.25 | 61.64 | 59.85 | 61.41 | 59.43 | 58.35 |
| Qwen2.5-VL | 7B | 57.29 | 57.45 | 67.40 | 67.35 | 63.60 | 62.87 | 58.23 | 58.34 |
| Intern-VL3 | 8B | 55.10 | 54.30 | 64.80 | 63.67 | 66.70 | 65.21 | 56.32 | 55.22 |
| **SCPO** | 7B | **65.80** | **65.21** | **74.60** | **74.52** | **80.75** | **76.98** | **67.15** | **66.83** |
| *Human* | | | | | | | | | |
| Human | - | 75.7 | 75.6 | 74.0 | 73.5 | 96.0 | 91.7 | 76.8 | 80.3 |

Qwen2.5-VL and InternVL3, to prove the effectiveness not from foundation model but our framework. For multi-agent baseline, we adopt the MMD-Agent framework using task decomposition and knowledge integration paradigms and compare it on MMFakeBench.

**Settings.** For fair comparison, we implement two prompt strategies. **Open-Prompting (main setting)**: The specific misinformation type is unknown with only *"You are a professional multimodal information analyzer..."* in system prompts. This setting explores the effectiveness and generalization in real situations. We use Qwen2.5-7B-Instruct to parse model predictions. **Closed-Prompting**: We clearly inform the misinformation type of each sample in system prompts to explore in-domain performances. Our model always uses open prompting to show its generalization capability.

## 4.2 MAIN RESULTS

Tab.1 presents the overall performance of SCPO and other leading MLLMs on MFC-Bench under the open-prompting setting [2].

**Comparison of Different MLLMs.** **Base Model**: Our SCPO framework significantly enhances the performance of its base model Qwen2-VL. SCPO achieves an overall accuracy of 67.15% and F1 score of 66.83%, marking a substantial improvement of 9.91% in accuracy and 9.92% in F1 score over Qwen2-VL. This demonstrates the remarkable effectiveness of our SCoT data and SCPO algorithm. **Stronger Open-Source Models**: Our 7B SCPO model not only surpasses its foundation but also outperforms stronger and larger open-source models. For instance, SCPO surpasses Qwen2.5-VL (58.23% accuracy) and 8B InternVL3 (56.32% accuracy), indicating internalizing social reasoning is more effective than simply enhancing the foundation. **Proprietary Models**: SCPO achieves highly competitive results against advanced proprietary models. It surpasses Claude3.5-Sonnet (66.85% accuracy) and closely approaches GPT-4o (69.11% accuracy). This shows that our framework can elevate a 7B open-source model to a competitive level with SOTA proprietary systems, making advanced multimodal misinformation detection more accessible and reliable.

**Comparison across Different Subtasks.** SCPO exhibits robust performance across all three distinct subtasks on MFC-Bench: Manipulation, OOC, and Veracity classification. We achieve the best among all open-source models and also competitive with proprietary models. For example, we

---

[2]The closed-prompting comparison is in Appendix Tab.6, and evaluation details are in Appendix A.2.3.

Table 2: Comparison with different models on MMFakeBench test set with standard prompting (Standard) and MMD-Agent framework. **Top-1** indicates the selection of the most likely answer labels, and **Recall** indicates the extraction of all involved predicted labels from model responses to detect whether the ground-truth is included. The accuracy, macro F1 score(%), precision and recall are reported as the metrics. The results marked by * are reproduced.

| Models | Size | Answer Generation | Test | | | |
|---|---|---|---|---|---|---|
| | | | F1 | Precision | Recall | Accuracy |
| *Proprietary Models (Close Prompting)* | | | | | | |
| GPT-4V | - | Standard | 54.0 | 62.1 | 48.8 | 61.5 |
| | | MMD-Agent | 67.7 | 63.0 | 48.7 | 59.1 |
| *Open-Source Models (Closed Prompting)* | | | | | | |
| Qwen-VL | 7B | Standard | 25.0 | 30.0 | 7.5 | 31.5 |
| LLaVA-Next | 7B | Standard | 19.0 | 16.5 | 26.9 | 32.3 |
| Qwen2-VL | 7B | Standard* | 31.8 | 36.8 | 41.2 | 41.2 |
| | | MMD-Agent* | 32.8 | 46.9 | 35.7 | 35.6 |
| LLaVA-Next | 13B | Standard | 41.0 | 40.6 | 25.0 | 37.4 |
| | | MMD-Agent | 40.6 | 42.7 | 31.2 | 37.5 |
| LLaVA-Next | 34B | Standard | 33.7 | 48.7 | 25.7 | 46.6 |
| | | MMD-Agent | 48.7 | 44.1 | 49.6 | 40.5 |
| *Open-Source Models (Open Prompting)* | | | | | | |
| Qwen2-VL | 7B | LLM-Parsing (Top-1) | 37.8 | 49.2 | 42.9 | 41.0 |
| | | LLM-Parsing (Recall) | 49.2 | 62.6 | 50.7 | 50.8 |
| **SCPO** | 7B | LLM-Parsing (Top-1) | **57.3** | **58.0** | **62.4** | **59.2** |
| | | LLM-Parsing (Recall) | **66.1** | **68.5** | **70.1** | **66.0** |

Table 3: Comparison with other LLM technologies using the same SCoT data based on Qwen2-VL.

| Models | Size | Manipulation | | OOC | | Veracity | | Overall | |
|---|---|---|---|---|---|---|---|---|---|
| | | Accuracy | F1 | Accuracy | F1 | Accuracy | F1 | Accuracy | F1 |
| Qwen2-VL | 7B | 57.49 | 57.09 | 63.85 | 63.84 | 50.55 | 54.21 | 57.46 | 57.09 |
| Self-Consistency | 7B | 62.83 | 60.44 | 52.60 | 52.58 | 52.20 | 55.56 | 61.63 | 58.35 |
| SFT | 7B | 63.14 | 61.51 | 67.65 | 67.03 | 77.20 | 75.33 | 64.20 | 63.10 |
| SFT+DPO | 7B | 55.97 | 54.44 | 68.75 | 67.64 | 78.45 | 71.39 | 57.98 | 56.09 |
| ORPO | 7B | 64.81 | 64.27 | 74.70 | 74.67 | 80.90 | 77.45 | 66.30 | 66.01 |
| **SCPO** | 7B | 65.80 | 65.21 | 74.60 | 74.52 | 80.75 | 76.98 | **67.15** | **66.83** |

surpass GPT-4o (69.85% accuracy) and Claude3.5-Sonnet (80.68% accuracy) for veracity classification. These results indicate the self-distilled collective cognition equips MLLMs with a comprehensive and nuanced understanding of different misinformation types.

**Comparison with Multi-Agent Baselines.** Tab.2 presents a comprehensive comparison on MM-FakeBench, where we focus our analysis on the performance of the MMD-Agent framework. The performance of MMD-Agent reveals that its agent-based decomposition strategy is only partially effective. Although MMD-Agent improve F1 scores mostly, when applied to Qwen2-VL and 34B LLaVA-Next, it lowers the accuracy 5.6% and 6.1%, respectively. This suggests that while agent-based reasoning can structurally identify potential misinformation signals, its multi-step process may introduce cumulative errors or flawed logic.

In contrast, our SCPO model demonstrates consistently superior performance across all metrics. Notably, our 7B SCPO model (59.2% accuracy) substantially outperforms the MMD-Agent applied to the much larger 34B LLaVA-NeXT (40.5% accuracy). This result illustrates that our social self-distillation paradigm provides a more robust and effective path to advanced reasoning than simply applying a complex inference-time agentic framework to a larger MLLM.

## 4.3 EXPERIMENTAL ANALYSIS

**Effectiveness of SCPO.** To validate our SCPO framework, we conduct an ablation study with other widely-used prompting and fine-tuning strategies. As shown in Tab.3, all fine-tuning methods utilize the same high-quality social reasoning data curated by our pipeline for fair comparison.

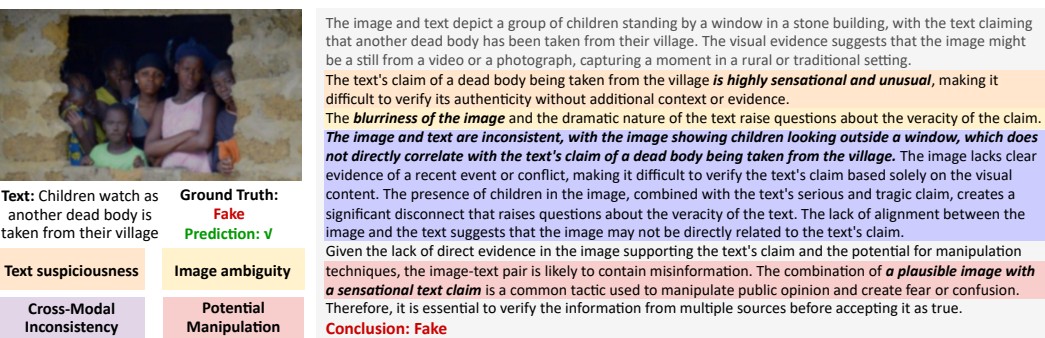

Figure 4: Visualization of our SCPO model on MFC-Bench.

Table 4: Comparison of the reasoning quality judged by GPT-4 on MFC-Bench.

| Models | Size | Misleadingness ↓ | Informativeness ↑ | Soundness ↑ | Readability ↑ |
|---|---|---|---|---|---|
| LLaVA-NeXT | 7B | 3.82 | 2.96 | 3.30 | 4.39 |
| | 13B | 3.61 | 3.07 | 3.48 | 4.49 |
| Qwen2-VL | 7B | 2.82 | 3.34 | 4.03 | 4.86 |
| **SCPO** | 7B | **2.51** | **3.69** | **4.19** | **4.93** |

**Value of Social Chain-of-Thought Data.** We first observe that standard Supervised Fine-Tuning (SFT) on the positive SCoT data establishes a strong baseline, achieving an overall accuracy of 64.20%. This confirms the high quality and effective pattern of our synthesized social reasoning data. The SFT results also achieve performance gains against self-consistency prompting strategy (61.63% accuracy), indicating the effectiveness of integrating diverse social perspectives. Preference optimization methods like ORPO, which integrate SFT with a preference loss term, further improve performance to 66.30% accuracy, showing advantages of learning to distinguish between high-quality and flawed reasoning paths.

**Superiority of Socially-Driven Scaling Signals.** Our SCPO model consistently outperforms all other methods, demontrasting the advantage of social correction value-driven scaling mechanism. Through verifiable social signals, our model focuses its efforts on resolving the most complex cognitive gaps, providing more robust and stable optimization.

**Model Interpretability.** A reliable misinformation detection model must provide convincing evidence in reasoning process. We evaluate the reasoning quality following the GPT-4 evaluation provided by MFC-Bench. The results in Tab.4 include four dimensions: *Misleadingness*, *Informativeness*, *Soundness*, and *Readability*, and SCPO achieves the best performance across all four metrics: The lowest *misleadingness* (2.51) indicates less logical mistakes or deceptive arguments. The highest on *informativeness* (3.69) and *soundness* (4.19) confirms the reasoning is logically coherent and well-supported by relevant multimodal evidence. The *readability* (4.93) shows the explanations are clear, well-structured and easily understandable. This superior interpretability is a direct result of our social self-distillation strategy, which integrates diverse analytical perspectives into the model's reasoning process, thereby enhancing both the quality and transparency [3].

**Visualization.** We provide a qualitative illustration of multi-perspective reasoning of our SCPO model in Fig.4. The image depicts children looking out of a window, with textual claim *children watch as another dead body is taken from their village*. Our generated reasoning reveals a detailed and traceable analytical process. It first identifies the *textual suspiciousness* of the unverifiable claim and *image ambiguity* due to blurriness. Crucially, it pinpoints the core *cross-modal inconsistency*, noting that the visual evidence does not correlate with the text narrative of a dead body. The model even demonstrates a deeper understanding by identifying the *potential manipulation* tactic of pairing

---

[3]More experiment results are detailed in Appendix A.4.

a plausible image with sensational text. Finally, by synthesizing these diverse reasoning paths, our model confidently reaches the correct conclusion: Fake. Through this sample, we illustrate how our social self-distillation equips MLLMs with diverse analytical perspectives as comprehensive human-like thinking, thus effectively detecting complex multimodal misinformation.

## 5 CONCLUSION

In this work, we introduce a multi-social-agent self-distillation framework, resolving the critical trade-off between the limited perspective of single-agent models and inefficiency of multi-agent systems for MMD task. In our framework, we first generate Social Chain-of-Thought (SCoT) data to distill diverse social perspectives into high-quality training examples. Based on SCoT data, we propose the Social Correction Value-Driven Preference Optimization (SCPO) algorithm, which uniquely leverages the degree of social misjudgement as a verifiable signal to adjust training. Our extensive experiments demonstrate the effectiveness, with our 7B model surpassing larger open-source models, dedicated multi-agent frameworks and even powerful proprietary models on challenging misinformation benchmarks. Ultimately, our work provides insights for distilling collective social intelligence into foundation models, offering a robust and efficient path to thinking as society.

## 6 ACKNOWLEDGMENTS

This work is supported by the National Natural Science Foundation of China under Grants 62425307, 62472303, and 62402335.

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
