# A   APPENDIX

This appendix contains additional details for the ICLR 2026 publication, titled "Thinking as Society: Multi-Social-Agent Self-Distillation for Multimodal Misinformation Detection". The appendix is organized as follows:

## A.1   DERIVATION OF THE FINITENESS OF SOCIAL CORRECTION VALUE

In this section, we provide a formal proof that the social correction value $sc(x)$ is bounded within the range $[0, 1]$. This property is crucial for the stability of the SCPO loss function during training.

First, we restate the formula for the social correction value function $sc(x)$:

$$sc(x) = 1 - \left( \frac{N_C}{N} + \frac{N_P}{N - N_C} \cdot \frac{1}{N} \right)$$

The variables are defined as follows:

- $N$: The **total number** of sampled users for a given multimodal sample $x$. $N$ is a positive integer such that $N > 0$.
- $N_C$: The number of users in the **correct set**. $N_C$ is an integer such that $0 \leq N_C \leq N$.
- $N_P$: The number of users in the **partially-correct set**.

The constraint of **partially-correct set**: A subset of the users who are *not* in the correct set. Therefore, the number of partially correct users must be less than or equal to $N - N_C$.

We will analyze the formula by considering two boundary cases for $N_C$: when $N_C < N$ and when $N_C = N$.

**Proof for the Upper Bound:** $sc(x) \leq 1$.   To prove that $sc(x) \leq 1$, we must show that the term being subtracted from 1 is non-negative. Let $V(x)$ be this term:

$$V(x) = \frac{N_C}{N} + \frac{N_P}{N - N_C} \cdot \frac{1}{N}$$

Based on our constraints:

- Since $0 \leq N_C \leq N$, the term $\frac{N_C}{N}$ is always non-negative.
- Since $0 \leq N_P$ and $N - N_C \geq 0$, the term $\frac{N_P}{N-N_C}$ is also non-negative.
- Since $N$ is positive, $\frac{1}{N}$ is positive.

As $V(x)$ is a sum of non-negative terms, $V(x) \geq 0$. Therefore:

$$sc(x) = 1 - V(x) \leq 1$$

**Proof for the Lower Bound:** $sc(x) \geq 0$.  To prove that $sc(x) \geq 0$, we must show that $V(x) \leq 1$.

**Case 1:** $N_C < N$ (i.e., not all users are correct)

In this case, we start with our fundamental constraint: $N_P \leq N - N_C \rightarrow \frac{N_P}{N-N_C} \leq 1$.

We can create an upper bound for $V(x)$ by substituting the maximum possible value for the $\frac{N_P}{N-N_C}$ term, which is 1.

$$V(x) = \frac{N_C}{N} + \frac{N_P}{N - N_C} \cdot \frac{1}{N} \leq \frac{N_C}{N} + (1) \cdot \frac{1}{N}$$

$$V(x) \leq \frac{N_C + 1}{N}$$

Since $N_C$ is strictly less than N in this case, $N_C$ can be at most $N - 1$.

$$N_C \leq N - 1 \implies N_C + 1 \leq N$$

Therefore:

$$\frac{N_C + 1}{N} \leq 1$$

This shows that $V(x) \leq 1$. From this, it follows that:

$$sc(x) = 1 - V(x) \geq 0$$

**Case 2:** $N_C = N$ (i.e., all users are correct)

In this case, the set of users who are not completely correct is empty. This implies that the set of partially correct users must also be empty, so $N_P = 0$. The formula for $sc(x)$ becomes:

$$sc(x) = 1 - \frac{N_C}{N} = 0$$

Thus, when all users perform correct reasoning, the social correction value is 0, which is consistent with the lower bound.

**Conclusion: Social correction value function $sc(x) \in [0, 1]$.**

## A.2 DATA DETAILS

### A.2.1 MULTI-SOURCE DATA CURATION

Multimodal Large Language Models (MLLMs) are pretrained on massive image-text data, exhibiting remarkable zero-shot generalization capabilities. Therefore, the training paradigms for many multimodal tasks have shifted to collecting high-quality downstream data for instruction tuning. For this paradiam, the diversity and balance of training data are important for the training stability and generalization ability of MLLMs. In this section, we introduce the multimodal misinformation dataset curation strategy in detail.

**Data Sources.**    Recent benchmarks cover multiple types of multimodal misinformation, including content manipulation of visual or textual modality, out of context misalignment, lack of factual veracity and fake news narratives. To ensure the diversity and representativeness of training samples, we curate a comprehensive dataset by aggregating multiple datasets spanning both synthetic and real-world multimodal misinformation:

**Synthetic Multimodal Misinformation Datasets (NewsCLIPpings, DGM4, AutoSplice):** These datasets are generated via controlled manipulation of text and images, enabling the model to learn explicit patterns of intentional forgery. NewsCLIPpings is an automatically generated out-of-context multimodal dataset containing both pristine and falsified instances, where falsified instances are un-manipulated but mismatched image-caption pairs. The NewsCLIPpings dataset consists of 71,072 samples in the training set and 7,024 in the validation set. DGM4 uses diverse image and text manipulation techniques (e.g., face swap, emotion manipulation, name entity replacement) to pristine instances from news data to generate misinformation automatically. The DGM4 dataset consists of 208,184 samples in the training set and 22,126 in the validation set. Autosplice is a manipulated image dataset that utilizes the text-to-image model DALLE-2 Ramesh et al. (2022) to edit images based on modified text. The Autosplice dataset consists of 3,420 samples.

**Real-world Fake News on Social Media (FineFake):**    These datasets are collected from actual social media platforms or news outlets, reflecting the complexity and subtlety of naturally occurring misinformation. FineFake is a novel multi-domain knowledge-enhanced fake news benchmark with fine-grained annotations, spanning different platforms like CNN and NewYork Times and covering diverse topics such as politics and business. The FineFake dataset consists of 16,909 samples.

**Semantic-Distributional Data Filtering.**    Given the substantial scale of the different datasets, directly utilizing them for training MLLMs is impractical due to potential redundancy, noise and inefficient resource allocation. Previous work discusses the method to analyze the feature distribution of multimodal datasets Zeng et al. (2024a). Following this setting, we filter each dataset by semantic and distributional similarity analysis, selecting a subset of samples that best captures its intrinsic characteristics. Specifically, our filtering process involves two core steps:

**Multimodal Semantic Sampling:** For each image-text pair, we extract CLIP embeddings for both visual and textual modalities. These image and text embeddings are fed into average pooling to generate a unified multimodal representation. We compute the cosine similarity between multimodal representations of training and validation set. The training samples are subsequently ranked in descending order based on their average CLIP similarity to the validation set, ensuring that top-ranked samples exhibit stronger semantic alignment with the evaluation distribution.

**Distributional Sampling:** Complementing the semantic analysis, we assess the distributional consistency between training and validation set using the negative calibrated Wasserstein gradient. This metric quantifies the discrepancy in gradient distributions between the two sets, providing a robust measure of how well the training subset mirrors the underlying distribution of the validation data. We also sort the training samples in descending order based on this gradient metric to minimize the distribution gap.

To ensure the preservation of core dataset characteristics while reducing redundancy, we adopt a dual-filtering strategy: for each source dataset, we retain samples that rank highly in both sorted lists (i.e., top performers in both semantic similarity and distributional consistency). For the datasets which lack official validation sets, we employ an unsupervised variant of multimodal semantic sampling, which uses KMeans clustering on CLIP embeddings to ensure both representativeness and diversity. Finally, we perform label balance verification on the filtered subset, ensuring an approximately equal proportion of real and fake samples. This step eliminates potential label bias for training stability. Through this process, we obtained a high-quality training dataset consisting of 9,773 samples.

**User Profile Database.**    For reliable social simulation, we construct user agents based on profiles with diverse demographic, psychological, and interest-based attributes. To this end, we utilize user data from the OASIS social simulator to model social interactions with realistic agent behaviors. The profile fields are shown in Tab.5.

Table 5: The field name and corresponding description in user profiles from OASIS platform.

| Field Name | Description |
|---|---|
| realname | Records the user's real full name to maintain authenticity of the user profile. |
| username | Serves as a unique identifier for the user in social simulation scenarios. |
| bio | Provides a concise self-introduction of the user, reflecting their core identity, hobbies, or life attitudes. |
| persona | Offers detailed background information about the user, guiding role-consistent feedback. |
| age | Specifies the user's age to reflect age-related characteristics. |
| gender | Notes the user's gender, which may influence certain viewpoint tendencies. |
| mbti | Indicates the user's personality type, shaping their thinking style and reasoning patterns. |
| country | Identifies the user's national background, accounting for regional cultural differences. |
| profession | Describes the user's professional field, enabling role-specific analysis. |
| interested_topics | Lists topics the user cares about, supporting topic-driven user sampling for relevant discussion. |

In total, we curate a user pool of 1,000 profiles, ensuring sufficient diversity in demographics (e.g., global representation across countries), professions (e.g., journalists, engineers, educators), and interest distributions.

### A.2.2 BENCHMARK ANALYSIS

**MFC-Bench.** MFC-Bench is a comprehensive benchmark specifically designed to evaluate the factual accuracy and trustworthiness of MLLMs in the context of fact-checking. The benchmark is organized into three key stages of verdict prediction, covering *35K* multimodal samples sourced from a diverse range of real-world scenarios and manipulated datasets. Its structured approach allows for a granular assessment of different reasoning capabilities.

- **Manipulation Classification:** This is the most challenging task within the benchmark, designed to evaluate MLLMs to identify whether multimodal content has been fabricated or altered. It includes a wide variety of sophisticated manipulation types, such as face swapping, text style transfer, face attribute editing, background changes, and AI-generated content. This task requires not just surface-level analysis but also deep background knowledge and sophisticated reasoning to detect subtle artificial artifacts.

- **Out-of-Context (OOC) Classification:** This task focuses on a common form of misinformation where both the image and text are individually authentic but are combined in a misleading way. It evaluates a model's ability to discern contextual coherence and semantic alignment between the visual and textual modalities, a crucial skill for identifying misleading multimodal content.

- **Veracity Classification:** This task assesses the model's ability to use an image as visual evidence to either support or refute a given textual claim. It is the multimodal equivalent of standard fact-checking and relies on the MLLM internal knowledge to verify the relationship between the presented textual claim and visual information.

We select MFC-Bench for several key reasons. First, its three-part structure provides a holistic and challenging evaluation that goes beyond simple classification, allowing us to evaluate the nuanced multi-perspective reasoning fostered by our SCoT data. Second, MFC-Bench is challenging because it shows even SOTA proprietary models have significant improvement space, making it an ideal platform to demonstrate the performance gains from our SCPO framework. Finally, MFC-Bench focuses on MLLM-centric justification quality, aligning with our goal of enhancing not just the accuracy but also interpretability of misinformation detection models.

**MMFakeBench.** MMFakeBench is the first comprehensive benchmark designed to evaluate multimodal misinformation detection in realistic mixed-source scenarios. It addresses a critical limitation of previous datasets, which often assume a single known source of forgery for each sample (e.g., text-only or vision-only manipulation). MMFakeBench indicates that real-world misinformation often stems from multiple forgery types, and its mixed-source detection requires creating robust and general-purpose detectors.

MMFakeBench is structured around three primary misinformation sources, which are further divided into 12 distinct sub-categories:

- **Textual Veracity Distortion:** In these samples, the text is fabricated (comprising natural, artificial, or GPT-generated rumors), while the supporting image is either authentic or AI-generated to be highly relevant and convincing. This evaluate the detection ability to fact-check a textual claim even when the visual evidence appears to support it.

- **Visual Veracity Distortion:** In this case, the textual claim is factually correct, but the accompanying image has been manipulated (either with Photoshop or via AI generation) to include fact-conflicting elements. These samples evaluate the ability to spot visual inconsistencies or artifacts when the model is faced with truthful texts.

- **Cross-modal Consistency Distortion:** This category contains samples where both the text and the image are individually authentic and correct, but their combination may convey false information. The misinformation arises from their mismatched pairing, which creates misleading presentation. This task evaluates the ability to reason about the semantic and contextual relationship between modalities.

The mixed-source design of MMFakeBench provides a highly realistic and challenging evaluation for the generalization capabilities, forcing the model to distinguish between multiple potential forgery types rather than specializing in only one misinformation category. This setting makes it an ideal environment to validate the robust and multi-perspective reasoning instilled by our multi-social-agent self-distillation framework.

### A.2.3 EVALUATION PROMPTS OF BOTH EXPERIMENTAL SETTINGS

To ensure the reproducibility of our experimental results, this section details the exact prompts used to evaluate the MLLM-based baselines in both the open and closed prompting settings. For all prompts, *[image]* is a placeholder for the input image and *[text]* is a placeholder for the input textual claim.

**Open Prompting.** The open-prompting setting is our main evaluation setup, designed to simulate real-world scenarios where the specific type of misinformation is unknown. We employ a single and unified prompt across all tasks in both the MFC-Bench and MMFakeBench to evaluate the general-purpose detection capabilities.

---

Prompts in Open Setting

**System prompt:** You are a professional multimodal information analyzer. Your task is to analyze the provided image-text pair for potential misinformation.
**Task Prompt:** Please provide a detailed reasoning and final conclusion of the authenticity of the given multimodal contents. If you think the provided contents are fake, your reasoning should contain the specific elements that appear suspicious or manipulated.

Analyze this following image-text pair:

Image: *[image]*
Text: *[text]*

Your detailed reasoning and final conclusion:

---

**Closed Prompting.** The closed-prompting setting is used to evaluate the in-domain performance of baseline models on the specific subtasks. In this setting, the prompt explicitly informs the model of the specific misinformation type. Our experiments adopt the same prompts provided by MFC-Bench and MMFakeBench.

**MFC-Bench.**

---

**Prompts for Manipulation Classification**

**System Prompt:** Manipulation encompasses various alterations such as face swapping, face attribute editing, background changing, image generation, entity replacement, and style transfer. Your task is to determine if the image and caption have been manipulated.

**Task Prompt:** Given a claim *[text]* and its image *[image]*, is this multimodal content manipulated?

---

**Prompts for OOC Classification**

**System Prompt:** Out-of-Context Classification is a task in which the goal is to identify whether a given image and accompanying text are contextually mismatched or falsely connected. Your task is to identify whether a given image and its accompanying text are contextually mismatched or falsely connected.

**Task Prompt:** Does this claim *[text]* match its image *[image]* ?

---

**Prompts for Veracity Classification**

**System Prompt:** The Veracity task in a multimodal context involves assessing the truthfulness or accuracy of textual claims by using visual evidence. Your task is to determine the truthfulness of textual claims based on the accompanying visual evidence.

**Task Prompt:** Based on the image *[image]*, is this claim *[text]* true?

---

**MMFakeBench.**

---

**Standard Prompts for Three-Stage Detection**

*[image]*
Given a multimodal misinformation, it contains both news caption and news image. News caption is: *[text]*

To make a accurate judgement of the multimodal misinformation, please follow the instructions bellow:
1. Is there any credible objective evidence refuting the news caption? If yes, please answer in the form: 'Finish[TEXT REFUTES].'. If no, continue to step 2.
2. Is there any credible objective evidence refuting the news image? If yes, please answer in the form: 'Finish[IMAGE REFUTES].'. If no, continue to step 3.
3. Does the news caption match the content of news image? If yes, please answer in the form: 'Finish[ORIGINAL].'. If no, please answer in the form: 'Finish[MISMATCH].'.

You should answer in the following form: 'Finish[TEXT REFUTES].' or 'Finish[IMAGE REFUTES].' or 'Finish[ORIGINAL].' or 'Finish[MISMATCH].'. The answer is:

---

### A.2.4 ANSWER LABEL GENERATION

Because our SCPO model adopt open-ended reasoning, it makes traditional evaluation methods such as automated answer matching insufficient for our model. To ensure a consistent evaluation, we employ Qwen2.5-7B-Instruct as a dedicated judger to parse the final prediction label from the generated response. For all prompts, *[output]* is a placeholder for the model response.

**MFC-Bench.** The evaluation of MFC-Bench adopts binary classification, so the prompt for answer extraction is as follows:

Prompts for Answer Extraction on MFC-Bench

The following is an analysis of misinformation by an AI system:

Analysis:
*[output]*

This analysis is used to do binary classification of whether the given multimodal information is real or fake.

Please give the final conclusion of this analysis and ONLY answer "real" or "fake":

**MMFakeBench.** The evaluation of MMFakeBench adopts multi-classification. For multi-class settings, we provide Qwen2.5-7B-Instruct with the statements of each label, and design the following two answer extraction methods:

- **Top-1 Selection:** We prompt Qwen2.5-7B-Instruct to select the most likely answer label. This setting is the standard approach for multi-class classification.

- **Recall Evaluation:** We prompt Qwen2.5-7B-Instruct to return all potentially-involved answer labels in the model response. If the ground-truth label is included in the extracted labels, we consider the model to answer correctly. This setting is to explore the complete reasoning and judgement abilities.

The prompts for both settings are as follows:

Prompts for Top-1 Selection on MMFakeBench

The following is an analysis of misinformation by an AI system:

Analysis:
*[output]*

There are four types of the final conclusion:
**real:** The analysis thinks the given image-text pair is real
**textual_veracity_distortion:** The analysis thinks the image-text pair contains text-based incorrect claims or rumors
**visual_veracity_distortion:** The analysis thinks the misinformation is in the images
**mismatch:** The analysis thinks although the image and text are individually accurate, the combination of image and text creates potential misinterpretations due to incorrect associations or semantic discrepancies

**Please give the final answer of this analysis, select the clearest (top-1) answer in [real, textual_veracity_distortion, visual_veracity_distortion, mismatch], ONLY give the final choice:**

---

**Prompts for Recall Evaluation on MMFakeBench**

The following is an analysis of misinformation by an AI system:

Analysis:
*[output]*

There are four types of the final conclusion:
**real:** The analysis thinks the given image-text pair is real
**textual_veracity_distortion:** The analysis thinks the image-text pair contains text-based incorrect claims or rumors
**visual_veracity_distortion:** The analysis thinks the misinformation is in the images
**mismatch:** The analysis thinks although the image and text are individually accurate, the combination of image and text creates potential misinterpretations due to incorrect associations or semantic discrepancies

**Please give the final conclusion of this analysis, select all occurred answers in [real, textual_veracity_distortion, visual_veracity_distortion, mismatch], ONLY give the final list:**

---

### A.3 IMPLEMENTATION DETAILS

#### A.3.1 DETAILED ARCHITECTURE OF COORDINATOR AND SUMMARIZER AGENTS

Directly concatenating feedback from all users may have long-context challenge and affect the quality of the preference data synthesis. To address this, we design a hierarchical synthesis strategy for the coordinator and summarizer agents:

- First Level (Grouping by Profession): We first group the user responses (from the *potential correct* or *incorrect* set) into smaller sub-groups based on their professions. This design stems from the insight that profession plays a key role in shaping users' thinking patterns.
- Second Level (Hierarchical Synthesis): The coordinator and summarizer agents then process each professional sub-group (typically 2-5 responses) to generate initial integration. Finally, they synthesize these first-level summaries into a single, coherent, and comprehensive reasoning chain.

This hierarchical, profession-based approach allows the coordinator and summarizer agents to comprehensively incorporate the user attributes and effectively manage the long context of multi-perspective feedback, ensuring that diverse and unique insights are preserved and integrated.

#### A.3.2 TRAINING SETTINGS

We introduce twenty social users for each sample in our simulation because we observe that this scale can balance user diversity and simulation costs. In the training process, we adopt full-parameter fine-tuning of Qwen2-VL-7B-Instruct. We set the learning rate to 2e-5 and employ a cosine learning rate scheduler. To prevent initial instability, we use a warm-up phase of 100 steps. The batch size and gradient accumulation steps are both 4, and the training duration is approximately 8 hours. The SCPO model is trained for a total of 3 epochs, which we find to be sufficient for convergence without significant overfitting. All training is performed on 2 NVIDIA A800 80GB GPUs. We complete SCPO training using LLaMA Factory Zheng et al. (2024), a robust and widely-used open-source framework designed for the efficient fine-tuning of large language models.

### A.4 MORE EXPERIMENTAL RESULTS

#### A.4.1 DESCRIPTION OF MLLM-BASED BASELINES

To ensure comprehensive evaluation of our SCPO framework, we select diverse MLLM baselines. Our selection spans both SOTA proprietary and leading open-source models, covering a wide range of parameter scales and architectural designs.

**Baselines on MFC-Bench.**   We conduct extensive experiments on MFC-Bench to evaluate the following representative MLLM baselines.

**Proprietary Models:** These models represent the current SOTA in multimodal reasoning and serve as the primary benchmark for top performance.

- **GPT-4o (OpenAI):** As OpenAI's flagship multimodal model at the time of our experiments, GPT-4o is known for its powerful real-time reasoning capabilities across vision and text. It is included to establish the highest performance ceiling on the benchmark.

- **Claude 3.5 Sonnet (Anthropic):** A powerful model from Anthropic, particularly recognized for its strong visual reasoning and graphic analysis capabilities. It serves as another top-tier proprietary baseline to ensure our comparison is not limited to a single provider.

**Open-Source Models:** This category includes a wide range of publicly available models, allowing us to evaluate the performance of our framework relative to the broader research community across different scales and families.

**Large-Scale Models:**

- **InternVL (25.5B):** A powerful vision-language foundation model, included to represent the performance of larger-scale open-source models.

- **CogVLM (17B):** A well-regarded open-source model known for its deep fusion of vision and language features with a trainable visual expert module, providing another strong large-scale baseline.

- **LLaVA-NeXT (13B):** A leading model from the popular LLaVA series, selected for its strong reasoning and instruction-following capabilities in the medium-to-large parameter range.

- **Pixtral (12B):** [4] A specialized model from Mistral AI, known for its proficiency in understanding charts, figures and documents, making it a relevant baseline for fact-checking.

**Stronger Similar-Scale Models:**   To demonstrate that the performance gains from SCPO are not simply due to using a better base model, we explicitly include MLLMs that are considered stronger than our base model (Qwen2-VL) but are in a similar parameter scale.

- **Qwen2.5-VL (7B):** A more advanced version with enhanced cross-modal reasoning abilities in the Qwen series, included to provide a direct comparison and show that our contribution is more significant than an incremental model update.

- **InternVL2 (8B)** [5] **and InternVL3 (8B):** These models represent recent and powerful advancements in the 8B parameter range. Especially, InternVL3 adopts native multimodal pretraining on large amounts of high-quality task data, proving better cross-modal understanding and reasoning abilities. These models serves as additional strong baselines to validate the effectiveness of our training paradigm.

**Base Model:**

**Qwen2-VL (7B):** Our SCPO framework is built upon and fine-tuned from the Qwen2-VL model. We select Qwen2-VL for its strong open-source foundation, compact size, and balanced performance. All improvements reported for SCPO are measured directly against the original Qwen2-VL's performance, providing a clear and fair assessment of our contribution.

**Baselines on MMFakeBench.**   In addition to the MLLM baselines detailed in MFC-Bench, our evaluation on MMFakeBench incorporates a unique multi-agent baseline.

**MMD-Agent:** The primary and most significant baseline for MMFakeBench. MMD-Agent is a multi-agent inference-time framework that integrates the reasoning, action, and tool-use capabilities of MLLMs. It addresses the mixed-source challenge by decomposing the detection task into a

---

[4]https://mistral.ai/news/pixtral-12b/
[5]https://internvl.github.io/blog/2024-07-02-InternVL-2.0/

Table 6: Comparison between our Qwen2-VL-based self-distillation framework and other MLLM baselines on MFC-Bench in close-prompting setting. The accuracy and macro-averaged F1 score(%) are reported as the metrics. The best and second test results are in bold and underlined, respectively. These results are reported by MFC-Bench.

| Models | Size | Manipulation | | OOC | | Veracity | | Overall | |
|---|---|---|---|---|---|---|---|---|---|
| | | Accuracy | F1 | Accuracy | F1 | Accuracy | F1 | Accuracy | F1 |
| *Proprietary Models (Closed Prompting)* | | | | | | | | | |
| GPT-4o | - | 65.7 | 60.4 | 84.8 | 84.8 | 80.1 | 63.0 | **67.7** | **69.4** |
| GPT-4V | - | 58.4 | 50.2 | 75.8 | 75.2 | 77.4 | 60.0 | 60.6 | 61.8 |
| Claude3.5-Sonnet | - | 59.9 | 41.7 | 49.9 | 37.6 | 72.7 | 47.4 | 60.1 | 42.2 |
| Claude3-Haiku | - | 51.4 | 37.8 | 59.8 | 59.5 | 80.3 | 57.4 | 53.7 | 51.6 |
| Gemini-1.5-Pro | - | 64.2 | 61.6 | 80.2 | 80.1 | 79.6 | 56.6 | 66.1 | 66.1 |
| *Open-Source Models (Closed Prompting)* | | | | | | | | | |
| Emu2 | 37B | 38.7 | 33.0 | 51.9 | 51.1 | 70.0 | 52.6 | 41.4 | 45.6 |
| InternVL | 25.5B | 60.1 | 44.6 | 73.4 | 73.0 | 80.0 | 57.4 | 62.1 | 58.3 |
| CogVLM | 17B | 56.3 | 52.3 | 61.4 | 56.2 | 76.4 | 63.4 | 57.8 | 57.3 |
| LLaVA-NeXT | 13B | 62.5 | 56.5 | 61.8 | 57.2 | 78.4 | 51.3 | 63.4 | 55.0 |
| InstructBLIP | 13B | 41.7 | 30.5 | 59.5 | 52.3 | 49.6 | 49.3 | 43.3 | 44.0 |
| Pixtral | 12B | 58.5 | 43.9 | 64.8 | 63.5 | 80.9 | 65.0 | 60.2 | 57.5 |
| MiniCPM-V-2.6 | 8B | 58.9 | 39.7 | 71.2 | 71.0 | 80.4 | 65.1 | 60.9 | 58.6 |
| LLaVA-OneVision | 7B | 61.5 | 55.5 | 75.7 | 75.4 | 80.9 | 60.3 | 63.5 | 63.7 |
| Molmo | 7B | 59.3 | 59.3 | 58.9 | 52.3 | 79.9 | 57.6 | 60.5 | 56.4 |
| Qwen-VL | 7B | 45.7 | 45.4 | 69.7 | 69.4 | 82.7 | 69.3 | 49.4 | 61.4 |
| Qwen2-VL | 7B | 59.9 | 46.6 | 80.1 | 80.1 | 85.7 | 75.5 | 62.7 | **67.4** |
| Yi-VL | 6B | 56.4 | 43.8 | 70.4 | 70.4 | 78.4 | 60.0 | 58.6 | 58.1 |
| xGen-MM | 5B | 42.7 | 33.8 | 50.0 | 44.8 | 64.7 | 48.7 | 44.5 | 42.4 |
| *Open-Source Models (Open Prompting)* | | | | | | | | | |
| Qwen2-VL | 7B | 57.4 | 57.0 | 63.8 | 63.8 | 48.9 | 52.8 | 57.2 | 56.9 |
| **SCPO** | 7B | 65.8 | 65.2 | 74.6 | 74.5 | 80.8 | 77.0 | **67.2** | 66.8 |

structured, three-stage pipeline: Textual Veracity Check, Visual Veracity Check, and Cross-Modal Consistency Check. At each stage, the MLLM-based agent uses its internal knowledge to generate reasoning process. Each agent also can use external tools such as the Wikipedia API to retrieve supplementary factual information.

### A.4.2   COMPARISON OF CLOSED-SETTING BASELINES ON MFC-BENCH

While our main evaluation focuses on the more challenging and realistic open-prompting setting, we also conduct a comprehensive evaluation in the closed-prompting setting to assess the in-domain performance of all baselines. In this setting, the model is explicitly prompted with the specific subtask it requires to perform (i.e., Manipulation, OOC, or Veracity classification). The detailed results of this evaluation are presented in Tab.6.

**Comparison of MLLM Baselines.**   Our SCPO framework demonstrate strong performance in this setting consistently. With an overall accuracy of 67.15% and an F1 score of 66.83%, SCPO is highly competitive with the best-performing proprietary model, GPT-4o (67.7% accuracy and 69.4% F1 score), and significantly outperforms other leading open-source models like InternVL (62.1% accuracy) and LLaVA-OneVision (63.5% accuracy).

**Comparison of Base Qwen2-VL.**   Even Qwen2-VL is informed with explicit misinformation type, our social self-distillation and preference optimization process can achieve F1 score improvement of 3.7% while maintaining a considerable accuracy. This suggests that our framework optimizes for a more balanced and robust reasoning capability across all tasks, improving the real-world misinformation detection. Overall, the results confirm that SCPO is a highly effective framework that excels in both challenging open-domain scenarios and more task-specific evaluations.

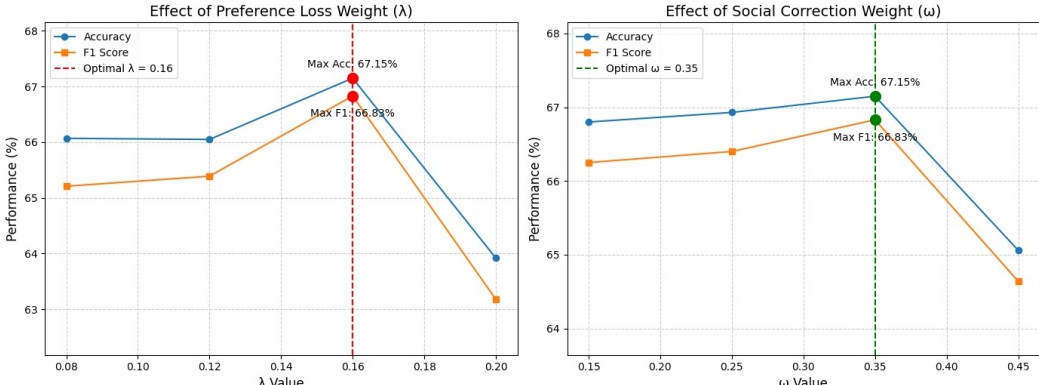

Figure 5: Ablation results of preference loss weight ($\lambda$) and social correction weight ($\omega$).

### A.4.3 ABLATION STUDIES OF KEY HYPERPARAMETERS IN SCPO

We conduct further ablation study on the two key hyperparameters of the SCPO loss function: the preference loss weight $\lambda$, and the social correction weight $\omega$. The results in Fig.5 validate our final hyperparameter selection.

**Effect of Preference Loss Weight ($\lambda$).** This hyperparameter controls the overall strength of the preference optimization term relative to the standard Supervised Fine-Tuning (SFT) loss. As shown in the left panel of Fig.5, when we assign a too low preference loss weight (e.g., 0.08), it results in suboptimal performance, as the model does not optimize enough from the SCoT preference pairs. Conversely, over-optimization based on high loss weight (e.g., 0.20) makes the model fit the preference data unexpectedly, leading to a degradation of its performance. We observe that the optimal balance is achieved at $\lambda = 0.16$, which yields the highest accuracy (67.15%) and F1 score (66.83%).

**Effect of Social Correction Weight ($\omega$).** This hyperparameter controls the strength of our novel social correction scaling mechanism. It determines how much to amplify the preference loss for samples with high social misjudgement. The right panel of Fig.5 shows that performance steadily increases as $\omega$ is raised from 0.15 to 0.35, confirming the effectiveness of our socially-driven signal. This trend indicates that prioritizing challenging samples with higher cognitive gaps leads to a more robust and capable model. However, beyond this point, increasing $\omega$ to 0.45 results in a performance decline. This suggests that an overly aggressive scaling can cause training instability by placing too much emphasis on a small subset of difficult samples. The best performance was achieved with $\omega = 0.35$, validating this value for our main experiments.

### A.4.4 ABLATION ANALYSIS FOR SOCIAL SIMULATION

In this section, we analyze the selected social simulation parameters (i.e., each sample has twenty social users participating in the discussion) to justify our choice.

**Experiment of User Diversity.** Each user profile contains the persona field, which specifically describes the detailed user information. We use MPNet [6] to encode all persona texts corresponding to a single sample, and calculate the average central semantic dispersion as follows:

$$e_c = \frac{1}{U} \sum_{i=1}^{U} e_i$$

$$sim_c = \frac{1}{U} \sum_{i=1}^{U} \frac{e_c \cdot e_i}{|e_c| \cdot |e_i|}$$

---

[6]https://huggingface.co/sentence-transformers/all-mpnet-base-v2

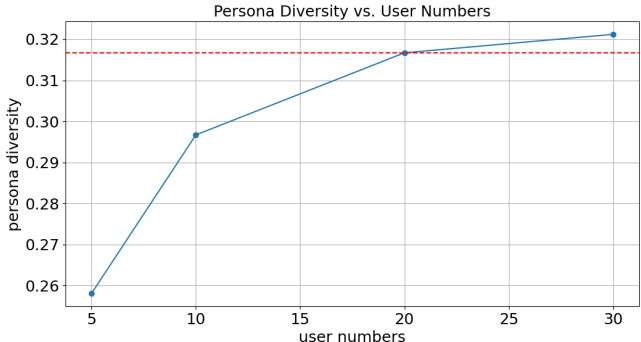

Figure 6: Curve between dispersion value and the number of sampled users.

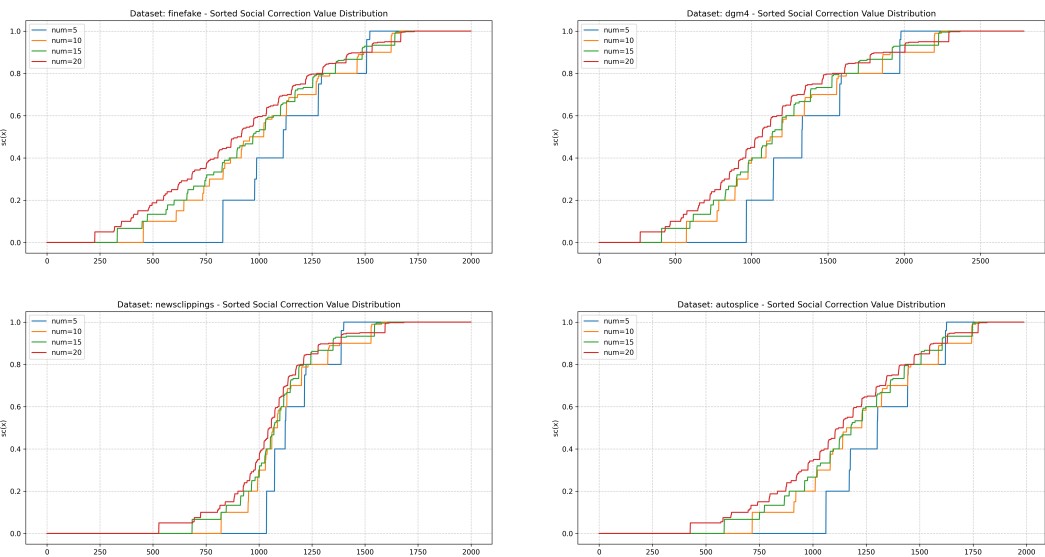

Figure 7: Curves of social correction value under different user numbers.

$$dis = 1 - sim_c$$

where $U$ denotes the user numbers for a sample, and $e_i$ denotes each persona embedding. This dispersion value represents the diversity of the sampled users. The stronger the user diversity, the broader the coverage of social simulation. We plot the relationship curve between dispersion value and the number of sampled users. As shown in Fig.6, We observe a significant marginal convergence at $U = 20$. The diversity improvement brought by further increasing the user numbers is very small and it is not worth additional inference costs for this group of users.

**Experiment of Social Correction Value Distribution.** Our social correction value (i.e., $sc(x)$) is the core signal which SCPO uses to prioritize challenging samples. Its distribution is a direct output of the social simulation. Therefore, we analyze how this distribution changes with the user numbers to provide direct insight into the quality of the training signal.

As shown in Fig.7, when the number of users is low like 5, the distribution of the social correction value has a relatively large deviation, indicating that insufficient participating users will make it difficult to fully exploit the training value of the sample. When the number of users gradually increases to 20, the distribution curve of social correction value becomes stable, indicating a more reliable and nuanced preference scaling signal.

**Proportion of Feedback Categories**   To ensure the training balance, we calculate the average proportion of users falling into each category (correct, partially correct, and incorrect) across the entire training dataset. The average distribution per sample is as follows:

Table 7: The average number of agents per sample in the training dataset.

| Feedback Category | Average Number of Agents per Sample (out of 20) | Percentage |
|---|---|---|
| Correct ($N_c$) | 9.5 | 47.49% |
| Partially Correct ($N_p$) | 0.5 | 2.28% |
| Incorrect | 10.0 | 50.23% |
| Total | 20.0 | 100% |

This distribution between correct and incorrect responses achieves balance within the simulated society, and the proportion of partially correct responses also contributes to the fine-grained term in the $sc(x)$ value. This indicates that the balanced distribution provides informative signals for our SCPO algorithm.

### A.4.5   INFERENCE COST ANALYSIS

**Data Generation Stage.**   In practice, we deploy Qwen2-VL-7B-Instruct using vLLM engine, and the generation of our full SCoT dataset (about 10,000 samples) required approximately 83.5 A800 GPU-hours. This involved querying 20 agents per sample, followed by our multi-stage classification, augmentation and synthesis pipeline. Although we use this resource for data generation, it represents a highly favorable trade-off for real-world applications. Through training on this SCoT dataset, we can obtain a model that performs advanced misinformation reasoning and detection in a single forward pass, facilitating practical deployment of our model.

**Inference Stage.**   To evaluate the practical efficiency of our SCPO framework for real-world applications, we conduct a detailed analysis of its inference cost, focusing on sample-level metrics such as latency (MLLM query times and average inference time) and comprehensiveness of reasoning (generated tokens). Both standard prompting and MMD-Agent adopt close prompting and generate a simple classification without detailed explanations. The results in Tab.8 compares our single-pass SCPO model against both standard prompting strategy and multi-step MMD-Agent framework.

The analysis reveals a significant efficiency advantage for our SCPO model, particularly when compared to the complex agentic approach. Due to the multi-step and decompositional nature, the MMD-Agent framework requires an average of 5.82 model queries per sample to complete its textual analysis, visual judgement, consistency checks and tool-use integration. This multi-query process introduces substantial latency and computational overhead. In contrast, our SCPO model operates in an end-to-end paradigm, requiring only a single query per sample to arrive at a final conclusion with clear reasoning process.

In terms of token generation, our SCPO model produces more tokens per query (215.03) than a standard baseline prompt (4.06). Our model generates multi-perspective reasoning, making it become more transparent and interpretable. It represents a highly favorable trade-off for overall efficiency. The MMD-Agent requires multiple sequential steps, leading to a much higher total inference time and a more complex deployment architecture.

Table 8: Comparison of inference cost on MMFakeBench. The number of MLLM queries, average inference time using vLLM engine and generated tokens for each sample are reported as metrics.

| Metrics (per sample) | Standard Prompting (Qwen2-VL) | MMD-Agent (Qwen2-VL) | Open Prompting (SCPO model) |
|---|---|---|---|
| query times | 1 | 5.819 | **1** |
| average inference time (vLLM) | 0.143s | 1.075s | **0.162s** |
| generated tokens | 4.06 | 153.17 | **215.03** |

This analysis demonstrates that our social self-distillation framework achieves superior accuracy with controllable inference cost. By distilling complex reasoning capabilities directly into models, SCPO avoids the high inference-time overhead associated with multi-agent systems, making it a more practical and scalable solution for real-world misinformation detection.

## A.5 MORE VISUALIZATION RESULTS

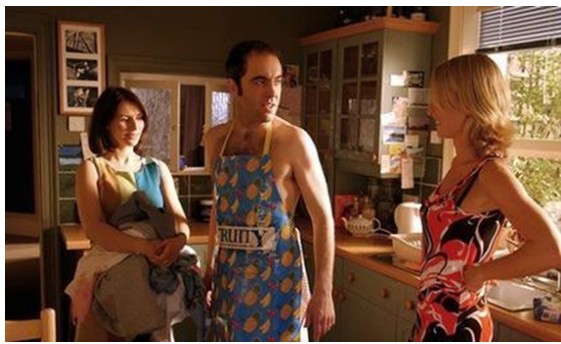

**Text:** James Nesbitt in a 2001 episode of Cold Feet which ran for five series on ITV

**Ground Truth: Real**

**Prediction: √**

The image and text pair depict a scene from the British sitcom "Cold Feet," featuring actors James Nesbitt, Harry Enfield, and Lisa Snowdon. ***The image shows the actors in a kitchen setting, wearing colorful aprons, which aligns with the visual style and characters typical of the show.*** The text mentions a 2001 episode of "Cold Feet," which aired for five series on ITV, and the visual and textual elements are consistent with the known context of the show. The attire and setting in the image are consistent with the visual style of the show, and the textual claims about the episode and series are plausible and align with known information. No signs of manipulation or inconsistencies were observed in either the image or the text. The alignment between the visual and textual elements, along with the absence of any apparent contradictions or manipulations, supports the authenticity of the content.
**Conclusion: Real**

Figure 8: Visualization for real sample of our SCPO model on MFC-Bench.

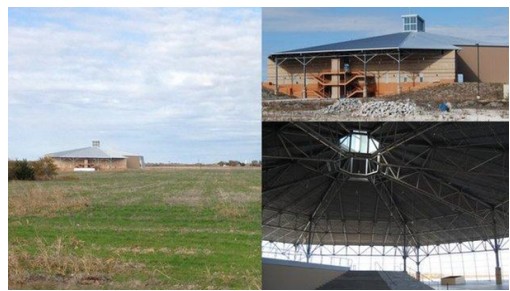

**Text:** The Redeemed Church plans to build a university sports complex and golf course on its land near Dallas

**Ground Truth: Fake**

**Prediction: √**

***The image and text do not align, raising significant concerns about the authenticity of the information.*** The image depicts a serene waterfront scene with people walking along a stone wall, while the text claims that The Redeemed Church plans to build a university sports complex and golf course on its land near Dallas. ***The visual evidence and textual claims are inconsistent, suggesting that the image may have been used to mislead viewers. The presence of a lighthouse in the background of the image, which is not mentioned in the text, further indicates that the image has been manipulated to create a false narrative.*** The lack of contextual alignment and the absence of visual evidence supporting the text's claims also contribute to the suspicion that the content is fabricated. The potential economic and social impacts of such a development, if it were to occur, are noteworthy. However, the lack of direct relevance to the fields of agriculture, food, natural resources, and economics makes it difficult to assess the credibility of the text's claims based solely on those perspectives. Given the inconsistencies and manipulations evident in the image and text, the content appears to be fabricated and designed to mislead viewers.
**Conclusion: Fake**

Figure 9: Visualization for out-of-context (OOC) sample of our SCPO model on MFC-Bench.

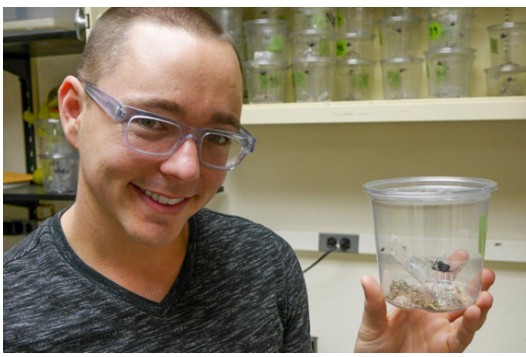

**Text:** Donald Trump tweeted that foreigners attending college in the United States who want to stay 'should not be thrown out of our country

**Ground Truth:** Fake

**Prediction:** √

The image depicts a person in a laboratory setting, holding a plastic container with some items, while the accompanying text discusses a tweet by Donald Trump regarding the stay status of foreigners attending college in the United States. The laboratory setting and the mention of a tweet about immigration policies do not align, raising significant doubts about the authenticity of the content. The tweet is found to be fake due to its false statements, misleading information, or content that does not conform to facts. The image, while potentially related to a scientific or research context, does not provide sufficient evidence to support the claims made in the text. Additionally, the tweet is classified as a statement of opinion rather than a fact, and its falsehoods and misleading information are highlighted by fact-checking websites. ***The content is also determined to be fake due to the presence of false statements, misleading information, or content that does not conform to facts. Given the lack of direct evidence linking the image to the text and the presence of misinformation, the content is considered fake.***
**Conclusion: Fake**

Figure 10: Visualization for veracity classification sample of our SCPO model on MFC-Bench.

### A.6    FAILURE ANALYSIS

In this section, to comprehensively understand existing limitations and guide future research, we present and analyze the following failure cases.

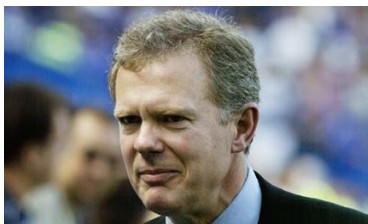

**Text:** Trevor Birch Portsmouth s second administrator in two years said he is used to dealing with clubs in crisis

**Ground Truth:** Real     **Prediction:** ×

***The image and text do not align, and there is no direct evidence linking the man in the image to the claim about Trevor Birch being the second administrator of Portsmouth Football Club.*** The image depicts a man at a sports event, while the text discusses the financial situation of a football club. This discrepancy raises significant doubts about the authenticity of the text. Additionally, the lack of contextual alignment between the image and the text makes it difficult to verify the credibility of the claim. The text appears to be plausible given the context of sports management and administration, but without additional verification, it is challenging to confirm its accuracy. ***The image, showing a man at a sports event, does not provide any visual evidence to support the text's claims, further suggesting that the image is not relevant to the text.***
**Conclusion: Fake**

Figure 11: Visualization for a failure case caused by overly cautious reasoning on MFC-Bench.

**Case 1: Overly Cautious Reasoning.**    As shown in Fig.11, the text identifies "Trevor Birch" as an administrator for a football club, and the image shows a man at a sports event. Our model correctly identifies the general scene (a man at a sports event) but concludes that the image and text do not align because there is no explicit visual evidence to confirm the man's identity as Trevor Birch. In this case, the model adopts an *overly cautious stance*, failing to recognize the visual evidence and give the "Fake" conclusion. This shows a limitation for handling visual evidence which is contextually relevant but not completely explicit.

**Case 2: Failure in Fine-Grained Entity Verification.**    As shown in Fig.12, the text names four specific actors from the film "Last Vegas", and the image shows four men who resemble the actors in a setting consistent with the film. The person on the far left is incomplete in the image, with only one arm.

Our model correctly identifies the general scene ("four men in a casual, beach-like setting") and evaluates the overall visual-textual consistency as high. However, it fails to perform the deeper, *fine-grained verification* required to confirm that the individuals in the photo are indeed the claimed actors. The model's reasoning is dominated by the strong surface-level consistency, leading it to an incorrect "Real" conclusion. This reveals a challenge in moving beyond general scene understanding to specific, real-world entity verification.

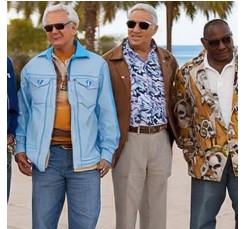

**Text:** Kevin Kline Morgan Freeman De Niro and Michael Douglas are four old friends who let loose in Las Vegas in the comedy Last Vegas

**Ground Truth: Fake**   **Prediction: ×**

*The image and text pair appear to be consistent and credible. The image depicts four men in casual attire, likely friends, standing together in a relaxed setting, possibly a beach or a pool area. The text identifies these individuals as Kevin Kline, Morgan Freeman, De Niro, and Michael Douglas, and mentions that they are four old friends who let loose in Las Vegas in the comedy "Last Vegas. " The visual evidence supports the authenticity of the image, showing a clear and well-lit scene with natural lighting and a beach-like backdrop. The attire and setting are consistent with a casual, relaxed environment, which aligns with the context of a comedy film. The textual claims are plausible, as the names and setting mentioned in the text are consistent with the visual elements. The image and text are also consistent, as the scene depicted in the image aligns with the narrative provided in the text.*

***From a professional standpoint, the image and text are evaluated based on their alignment with the film "Last Vegas," the plausibility of the claims, and the absence of obvious signs of manipulation.*** *The image and text are consistent and credible, with no signs of manipulation or inconsistencies noted. The content is consistent with the film "Last Vegas," and the visual and textual elements are aligned, supporting the claim that the image is likely a still from the film. The visual and textual elements are consistent with each other and with the film "Last Vegas."*

**Conclusion: Real**

Figure 12: Visualization for a failure case caused by lack of fine-grained entity verification on MFC-Bench.

## A.7 OTHER IMPORTANT PROMPTS

---

**Prompts for Topic Analysis**

**System Prompt:** You are a multimodal information analyzer.

**Task Prompt:** *[image]*
Given an image and attached text description, analyze which of the following topics are related based on its content:
Economics: Production, distribution, consumption of goods and services
Information Technology: Computers, networking, data management in business contexts
Culture & Society: Cultural expressions and societal structures
General News: Current events across politics, business, technology
Politics: Governance and power dynamics
Business: Commerce, trade, and business management
Fun: Leisure activities and entertainment

Text description:
*[text]*

Let's think step by step and give the related topics after the keyword "Final answer:".
Your thinking:

---

**Prompts for User Simulation**

**System Prompt:** You are a social media analyst simulating the perspective of a specific user. Your task is to analyze this image-text pair for potential misinformation while strictly adhering to your assigned role's characteristics.

**Role Profile**:
*[role profile]*

**Analysis Framework**
**Content Interpretation:**
- Describe how username would interpret both the image and text based on the user background
- Identify elements that would particularly stand out to this user

**Credibility Assessment:**
- Visual Evidence: Analyze image authenticity clues; Rationale: [explanation]
- Textual Claims: Evaluate claim plausibility; Rationale: [explanation]
- Contextual Alignment: Assess image-text consistency; Rationale: [explanation]

**Bias Analysis:**
- Potential confirmation biases based on interests
- Demographic/cultural influences on perception
- Professional expertise limitations

**Task Prompt:**
*[image]*
Multimodal Content for Analysis
*[text]*
Detection Task:
- Perform role-based analysis of potential misinformation:
    - Cross-validate image details with text claims
    - Identify contextual mismatches with your role-based and common knowledge
    - Identifying potential manipulation techniques
- Provide:
    - Role-specific risk assessment
    - Most/least convincing elements
    - Suggested verification methods this user would employ
- Conclusion Format: [As username's perspective] Verdict: [real/fake] Reasoning: [role-specific arguments]

Your thinking process:

---

---

**Prompts for Coordinator Agent**

**Task: Synthesize Role-based Multimodal Misinformation Analysis**

You are aggregating multiple analyses from different roles on an image-text pair into one complete reasoning and conclusion. Follow these steps:

**Content Synthesis:**

- Merge overlapping evidence while preserving all unique valuable insights (necessary and beneficial for getting correct conclusions) from different roles

- Remove all role-specific information such as user identifiers, professions and so on

- synthesize above contents into a fluent and detailed paragraph as complete reasoning

**Output Requirements:**

- Use professional misinformation detection and fact-checking tone

- After the fluent reasoning synthesizing all valid perspectives, end with a clear conclusion such as "Conclusion: [real/fake] (must match the ground truth)"

**Responses to Integrate:** *[output]*

**The synthesized content:**

---

**LLM Usage:** We use Gemini-2.5-pro [7] in paper writing for polishing statement or description. The specific operation is to manually complete the first draft of our manuscript, with Gemini-2.5-pro serving as an auxiliary refinement assistant.

---

[7]https://aistudio.google.com/