# OpenReview forum: "Thinking as Society: Multi-Social-Agent Self-Distillation for Multimodal Misinformation Detection"
_ICLR.cc/2026/Conference — ICLR 2026 Poster_

### Official Review · Reviewer_NiZd · 2025-10-28

**Soundness:** 2
**Presentation:** 2
**Contribution:** 2
**Rating:** 4
**Confidence:** 4

**Summary:**

This paper addresses multimodal misinformation detection (MMD) under realistic social media conditions. The authors argue that existing single-agent LMM-based detectors lack perspective diversity and are thus easily misled, while multi-agent systems provide multiple viewpoints but are inefficient and hard to optimize end-to-end. To resolve this trade-off, the paper proposes a “Multi-Social-Agent Self-Distillation” framework that aims to compress multi-agent social reasoning into a single model. Their SCPO-tuned 7B model achieves superior or comparable performance to larger open-source, multi-agent, and even proprietary models like GPT-4o and Claude 3.5-Sonnet on MFC-Bench and MMFakeBench under both open- and closed-prompt evaluations.

**Strengths:**

1. This paper attempts to model socially diverse reasoning and then distill it into a single deployable model is conceptually compelling and practically relevant for scalable moderation systems.
2. By coupling Social Chain-of-Thought (SCoT) data synthesis with the Social Correction Value-Driven Preference Optimization (SCPO) algorithm, the framework achieves robust reasoning performance.

**Weaknesses:**

1. The paper should be more precise here: is there any real agent interaction, disagreement resolution protocol, or iterative critique loop beyond single-shot persona prompting and post-hoc summarization? Otherwise, “multi-social-agent” risks overselling standard multi-persona sampling plus LLM judging.
2. Need for overlap analysis between source data and evaluation benchmarks.
3. The paper should clearly states: What is the size of the original pooled dataset (before filtering)? How large is the filtered “validation set”? How many instances survive as “High-quality Training Data” after selection?
4. The “Coordinator Agent” and “Summarizer Agent” are described as specialized components, yet their concrete implementation remains vague. Are these modules merely prompt-based LLM refiners, and if so, how do they differ functionally from standard LLM data post-processing?
5. Table 3 shows only marginal improvement of SCPO over ORPO (67.15 vs 66.30 accuracy; 66.83 vs 66.01 F1), suggesting that performance bottlenecks may stem more from the availability and quality of CoT-style supervision data than from the SCPO weighting mechanism itself

**Questions:**

6. The paper lacks clarity regarding human supervision and quality control, including the number of annotators involved, their expertise, and the specific labeling guidelines used to ensure annotation reliability.
7. The paper should report the proportion of samples assigned to each feedback category (correct, partially correct, incorrect), since these distributions directly influence the SCPO weighting and training balance.

---

> ### Author Response · Authors · 2025-11-21
> **Response to Reviewer NiZd (1)**
>
> We sincerely appreciate your insightful comments and valuable suggestions, and we carefully address each of your comments and provide point-by-point responses below.
>
> > Weakness 1: The paper should be more precise here: is there any real agent interaction, disagreement resolution protocol, or iterative critique loop beyond single-shot persona prompting and post-hoc summarization? Otherwise, “multi-social-agent” risks overselling standard multi-persona sampling plus LLM judging.
>
> **1. Precision of the Multi-Social-Agent Framework**
>
> We appreciate this important question, which prompts us to be more precise about our methodology. Our framework is intentionally designed to go beyond standard multi-persona sampling, and we would like to clarify the key distinctions:
>
> *   **Preserving Independent Perspectives**: The precondition of our self-distillation framework is to first elicit a wide range of **unbiased, independent social perspectives** without external influence. This allows us to capture a more authentic initial societal viewpoints. Introducing iterative debate or critique loops could lead to interference with each other, potentially masking the cognitive diversity we aim to distill.
>
> *   **Hierarchical User-Driven Synthesis**: The core of our multi-social-agent system lies in the preference data synthesis stage, which is far more than simple summarization. First, we clarify the working mechanisms of the coordinator and summarizer agents here, and we have supplemented the detailed architecture for data synthesis to the **Appendix A.3.1** (marked in blue). Specifically, our coordinator and summarizer agents employ a **hierarchical user-driven synthesis strategy**.
>  - **First Level (Grouping by Profession)**: We first group the user responses (from the `potential correct` or `incorrect` set) into smaller sub-groups based on their professions. This design is because profession is an important factor influencing the way of thinking for social users.
>  - **Second Level (Hierarchical Synthesis)**: The coordinator and summarizer agents then process each professional sub-group (typically 2-5 responses) to generate initial integration. Finally, they synthesize these first-level professional perspectives into a unified and comprehensive SCoT data.
>
>    This hierarchical, profession-based approach allows the coordinator and summarizer agents to comprehensively incorporate the user attributes and effectively manage the multi-perspective feedback, ensuring that diverse and unique insights are preserved and integrated. This hierarchical integration process is a key step that elevates our framework beyond simple sampling and judging.
>
> *   **Implicit Disagreement Resolution**: In fact, our **answer-centric feedback classification** implicitly handles disagreement. We group user agents into `potential correct` and `incorrect` sets based on their conclusions. The coordinator agent then focuses on synthesizing valuable perspectives from the `potential correct` set. This indicates that all users managed by the coordinator agent tend to reach the correct conclusion. The summarizer agent identifies the most representative flawed logic from the `incorrect` set. This indicates that all users managed by the summarizer agent tend to draw incorrect conclusions. Consequently, we can achieve internal consistency within the corresponding sets processed by the coordinator and summarizer agents. This separation and synthesis serve as an effective disagreement resolution mechanism for generating clear preference pairs.
>
> > Weakness 2: Need for overlap analysis between source data and evaluation benchmarks.
>
> **2. Overlap Between Source Data and Benchmarks**
>
> We confirm that there is **no overlap** between our training data and the evaluation benchmarks. Our training data is curated exclusively from the **training sets** of public datasets (NewsCLIPpings, DGM4, AutoSplice, and Finefake). The evaluation benchmarks, MFC-Bench and MMFakeBench, are specifically designed as held-out benchmarks for multimodal misinformation detection. This ensures a fair evaluation of all baselines and our models.

---

> ### Author Response · Authors · 2025-11-21
> **Response to Reviewer NiZd (2)**
>
> > Weakness 3: The paper should clearly states: What is the size of the original pooled dataset (before filtering)? How large is the filtered “validation set”? How many instances survive as “High-quality Training Data” after selection?
>
> **3. Size of Datasets During Curation**
>
> We appreciate the suggestion for these specific dataset size, and we have supplemented these detailed information about data curation to the **Appendix A.2.1** (marked in blue). The dataset sizes are as follows:
>
> *   **Original Pooled Dataset Size**:
>     *   NewsCLIPpings: 71,072 (train), 7,024 (validation)
>     *   DGM4: 208,184 (train), 22,126 (validation)
>     *   AutoSplice: 3,420 (train)
>     *   FineFake: 16,909 (train)
> *   **Filtering Strategy**:
>     *   For NewsCLIPpings and DGM4, we used their validation sets as references for our semantic and distributional sampling, as described in Appendix A.2.1.
>     *   For AutoSplice and FineFake, which lack official validation sets, we employ an unsupervised variant of semantic sampling, which uses KMeans clustering on CLIP features to ensure both representativeness and diversity.
> *   **Final High-Quality Training Data**: After our multi-stage filtering process and a final label balancing check, the resulting high-quality training dataset used for generating SCoT data consists of **9,773 instances**.
>
> > Weakness 4: The “Coordinator Agent” and “Summarizer Agent” are described as specialized components, yet their concrete implementation remains vague. Are these modules merely prompt-based LLM refiners, and if so, how do they differ functionally from standard LLM data post-processing?
>
> **4. Implementation of Coordinator/Summarizer Agents**
>
> We appreciate the suggestion for more clarity on the implementation of our Coordinator and Summarizer agents. As detailed in our response to Weakness 1, these agents are functionally distinct from standard post-processing because they employ a **specialized, hierarchical user-driven synthesis strategy** to handle the multi-perspective feedback. We first group users in `potential correct` or `incorrect` set by their professions, conduct intermediate integration for each professional group, and then synthesize these summaries. This multi-level process is crucial for effectively integrating and reasoning over the collective feedback. We have supplemented the detailed architecture for data synthesis to the **Appendix A.3.1** (marked in blue).
>
> > Weakness 5: Table 3 shows only marginal improvement of SCPO over ORPO (67.15 vs 66.30 accuracy; 66.83 vs 66.01 F1), suggesting that performance bottlenecks may stem more from the availability and quality of CoT-style supervision data than from the SCPO weighting mechanism itself
>
> **5. Improvement of SCPO**
>
> We appreciate the critical perspective on the performance gains. Regarding the performance gains over ORPO in Tab.3, we respectfully argue that on a large-scale benchmark like MFC-Bench (35K samples), the improvement of **~0.85% in accuracy and ~0.82% in F1-score is not insignificant**. This represents that the SCPO algorithm **has added nearly 300 successful judgments**. This indicates that both high-quality SCoT data and the SCPO algorithm play a significant role in the robust reasoning required for MMD task.

---

> ### Author Response · Authors · 2025-11-21
> **Response to Reviewer NiZd (3)**
>
> > Question 1: The paper lacks clarity regarding human supervision and quality control, including the number of annotators involved, their expertise, and the specific labeling guidelines used to ensure annotation reliability.
>
> **1. Human Supervision and Quality Control**
>
> Our framework is designed as a **self-distillation** pipeline, which intentionally minimizes the dependency for expensive human supervision during the data generation process.
>
> Moreover, to validate the quality and realism of our social simulation, **we immediately start a human evaluation study** to externally validate the realism of the agent-generated perspectives. We randomly sample a diverse set of user responses across various topics (e.g., politics, news, environment, biology) and professional domains (e.g., law, service industry, IT, public administration), and ask human evaluators to rate the consistency between the reasoning style and the corresponding user profile on a 5-point Likert scale. **Our preliminary results are highly promising**:
>
> | Level          | 1 (very inconsistent) | 2     | 3     | 4          | 5 (very consistent) |
> | :------------- | :-------------------- | :---- | :---- | :--------- | :------------------ |
> | **Percentage** | 0.71%                 | 2.14% | 8.57% | **49.29%** | **39.29%**          |
>
> These results indicate that **nearly 90% of the simulated responses are perceived by humans as consistent or very consistent with their assigned roles**, confirming that our simulation aligns well with human social cognition. This study validates the quality of our data without requiring human labeling. We will continue to gather more real investigation data for comprehensive evaluation of the actual diversity and realism of the generated reasoning.
>
> > Question 2: The paper should report the proportion of samples assigned to each feedback category (correct, partially correct, incorrect), since these distributions directly influence the SCPO weighting and training balance.
>
> **2. Proportion of Feedback Categories**
>
> We appreciate the suggestion for supplementing the proportion of feedback categories. For each sample, we categorize the user responses and then calculate the average proportion of users falling into each category across the entire training dataset. We have supplemented this proportion of feedback categories to **Appendix A.4.4** (marked in blue).
>
> The average distribution per sample is as follows:
>
> | Feedback Category            | Average Number of Agents per Sample (out of 20) | Percentage |
> | :--------------------------- | :---------------------------------------------- | :--------- |
> | **Correct** (`Nc`)           | 9.5                                             | 47.49%     |
> | **Partially Correct** (`Np`) | 0.5                                             | 2.28%      |
> | **Incorrect**                | 10.0                                            | 50.23%     |
> | **Total**                    | 20.0                                            | 100%       |
>
> This distribution between correct and incorrect responses achieves balance within the simulated society, and the proportion of partially correct responses also correspond to the fine-grained term in the `sc(x)` value. This indicates that the balanced distribution provides informative signals for our SCPO algorithm.
>
> Moreover, we have presented the sorted distribution curve of `sc(x)` (the red curve) in Fig.7 of Appendix A4.4. The steady rise of this `sc(x)` curve confirms that our dataset has balanced distribution of sample difficulties, from easy (low `sc(x)`) to hard (high `sc(x)`).  This ensures `sc(x)` can serve as a reliable and nuanced preference scaling signal.
>
> We hope these responses and new results have addressed your concerns. We are grateful for the opportunity to improve our paper based on your valuable feedback.

---

### Official Review · Reviewer_7qYS · 2025-10-30

**Soundness:** 3
**Presentation:** 4
**Contribution:** 3
**Rating:** 6
**Confidence:** 3

**Summary:**

This paper presents a Multi-Social-Agent Self-Distillation framework for multimodal misinformation detection (MMD) leveraging Large Multimodal Language Models (MLLMs). The proposed approach first simulates diverse social agents to produce multiview feedback on misinformation samples, synthesizing this into Social Chain-of-Thought (SCoT) data. Building on this, the Social Correction Value-Driven Preference Optimization (SCPO) algorithm is introduced, which dynamically scales the alignment loss by the observed social misjudgment. Experiments on MFC-Bench and MMFakeBench show that this self-distilled, socially-aware Qwen2-VL-based model outperforms a variety of open-source and proprietary baselines and matches or exceeds complex multi-agent or larger-scale systems on key multimodal reasoning tasks.

**Strengths:**

1. The paper tackles an important and timely challenge in MMD, seeking to integrate multi-perspective social reasoning into a single, efficient agent, thereby addressing the real-world complexity of misinformation and limitations of both single- and multi-agent systems.
2. The methodology is clearly articulated: simulating social context via user-profiled MLLMs, aggregating their reasoning, and synthesizing collective chains of thought. The data pipeline is well-detailed, and the synthesized dataset design is a practical advance.
3. SCPO introduces an interpretable, mathematically explicit mechanism to adjust model alignment based on task difficulty, which stands out against prior, less verifiable margin-scaling approaches.
4. The paper is competently written and mostly clear, with methodological details in both equations and pipeline diagrams.

**Weaknesses:**

1. While the diversity of simulated agents is claimed, the realism of their social perspectives is not externally validated. It remains unclear to what extent these agent-generated viewpoints transfer to true human social cognition. The manuscript omits ablation or validation against authentic crowd or expert responses, leaving the practical value of the “social” aspect somewhat speculative.
2. Although the authors proved the range of values ​​for $sc(x)$ in the appendix, I still have doubts about its simple form as $1+ωsc(x)$ in the social correction value: Is this linear weighting sufficient to distinguish samples of different difficulties? For example, for a completely correct simple sample, its social correction value is still 1, no different from the standard ORPO. Furthermore, the results in Table 3 show that the improvement of SCPO compared to ORPO is not significant, further raising concerns about the rationality of this weighting form.
3. During data synthesis, different large models naturally produce different responses. However, the paper does not specify which models are used for the user agents, coordinator agent, or summarizer agent, nor the rationale behind these choices. Moreover, model differences in handling long contexts (e.g., varying user_number) can substantially affect the outputs generated by the coordinator and summarizer agents, potentially introducing additional variability into the synthesized data.

**Questions:**

1. How well do the simulated social agents reflect real human social cognition? Specifically, is there any quantitative or qualitative validation comparing agent-generated perspectives with real crowd or expert responses?

2. Is the linear weighting $1+ωsc(x)$ sufficient and justified? Can the authors provide evidence that this formulation meaningfully differentiates sample difficulty, especially given the limited gains over ORPO in Table 3?

3. Which models are used for user agents, the coordinator, and the summarizer, and why? Do different context lengths (number of user numbers) affect the model selection for coordinator and summarizer agents?

**Details Of Ethics Concerns:**

N/A.

---

> ### Author Response · Authors · 2025-11-21
> **Response to Reviewer 7qYS (1)**
>
> We sincerely appreciate your insightful comments and valuable suggestions, and we carefully address each of your comments and provide point-by-point responses below.
>
> > Weakness 1: While the diversity of simulated agents is claimed, the realism of their social perspectives is not externally validated. It remains unclear to what extent these agent-generated viewpoints transfer to true human social cognition. The manuscript omits ablation or validation against authentic crowd or expert responses, leaving the practical value of the “social” aspect somewhat speculative.
> >
> > Question 1: How well do the simulated social agents reflect real human social cognition? Specifically, is there any quantitative or qualitative validation comparing agent-generated perspectives with real crowd or expert responses?
>
> **1. Realism of Simulated Social Perspectives (Weakness 1 & Question 1)**
>
> We appreciate the suggestion for validating whether our simulated agents can reflect true human social cognition.
>
> * **Human Validation:** To address this, **we immediately start a human evaluation study** to evaluate the realism of the agent-generated perspectives. We randomly sample a diverse set of user responses across various topics (e.g., politics, news, environment, biology) and professional domains (e.g., law, service industry, IT, public administration), and ask human evaluators to rate the consistency between the reasoning style and the corresponding user profile on a 5-point Likert scale. **Our preliminary results are highly promising**:
>
>   | Level          | 1 (very inconsistent) | 2     | 3     | 4          | 5 (very consistent) |
>   | :------------- | :-------------------- | :---- | :---- | :--------- | :------------------ |
>   | **Percentage** | 0.71%                 | 2.14% | 8.57% | **49.29%** | **39.29%**          |
>
>   These results indicate that **nearly 90% of the simulated responses are perceived by humans as consistent or very consistent with their assigned roles**. This provides evidence that our agent-generated viewpoints are not merely diverse but also realistic, effectively transferring to human social cognition. We will continue to gather more real investigation data for comprehensive evaluation of the actual diversity and realism of the generated reasoning.
>
> * **Foundation in Real-World Data**: Moreover, as detailed in Appendix A.2.1, our `User Profile Database` is constructed using data from the OASIS platform, which itself is based on information from real-world social media users. This ensures that the demographic, professional, and interest-based attributes of our simulated agents are grounded in reality.

---

> ### Author Response · Authors · 2025-11-21
> **Response to Reviewer 7qYS (2)**
>
> > Weakness 2: Although the authors proved the range of values for sc(x) in the appendix, I still have doubts about its simple form as `1+w*sc(x)` in the social correction value: Is this linear weighting sufficient to distinguish samples of different difficulties? For example, for a completely correct simple sample, its social correction value is still 1, no different from the standard ORPO. Furthermore, the results in Table 3 show that the improvement of SCPO compared to ORPO is not significant, further raising concerns about the rationality of this weighting form.
> >
> > Question 2: Is the linear weighting `1+w*sc(x)` sufficient and justified? Can the authors provide evidence that this formulation meaningfully differentiates sample difficulty, especially given the limited gains over ORPO in Table 3?
>
> **2. Justification and Sufficiency of the Linear Weighting `1 + w * sc(x)` (Weakness 2 & Question 2)**
>
> * **Design of Linear Weighting**: The linear weighting $1 + w * sc(x)$ is a deliberate design to **preserve the functional properties of `sc(x)`** while **ensuring training stability**. The bounded nature of $sc(x)\in [0,1]$ makes the scaling controllable (The detailed proof is located in Appendix A.1). Considering the weighting term before OR loss $\lambda(1+w*sc(x))$, OR loss is successfully scaled by the coefficients in the range of $[\lambda, \lambda(1+w)]$.
>
>   Moreover, this linear weighting approach to dynamically adjust loss function is adopted in other works like MM-DPO [1] (ICML 2025) and RLEV [2] (the latest work, arxiv 2510.20187). This shows our approach is aligned with state-of-the-art methodologies. In general, linear weighting is an effective method to preserve the properties of the underlying function and achieve controllable training.
>
> * **Differentiation of Sample Difficulty**: For a simple sample where all users are correct (`sc(x)=0`), the weighting term $1 + w * sc(x)$ becomes 1, and the loss term is equivalent to the standard ORPO loss. In fact, this is precisely the intended behavior. When a social consensus on the correct answer is already achieved, the current sample requires **no additional preference amplification**, allowing the model to maintain its general capability without overfitting.
>
>   In contrast, for a hard sample with high misjudgments (`sc(x)` approaches 1), the preference loss is amplified. This adaptive scaling mechanism allows SCPO to dynamically focus the training on challenging samples where social cognition is most valuable, rather than treating all samples equally like standard preference optimization algorithms.
>
> *   **Adaptive Scaling and Significance of Gains**: For more difficult samples where `sc(x) > 0`, our mechanism adaptively scales up the preference loss, forcing the model to focus more on resolving social misjudgment. This adaptive scaling is the core of our contribution. Regarding the performance gains over ORPO in Tab.3, we respectfully argue that on a large-scale benchmark like MFC-Bench (35K samples), the improvement of **~0.85% in accuracy and ~0.82% in F1-score is not insignificant**. This represents that the SCPO algorithm **has added nearly 300 successful judgments**, demonstrating that our socially-driven weighting effectively differentiates sample difficulty and improves overall reasoning capabilities.
>
> ---
> **References:**
>
> [1] Zhang et al. "MM-RLHF: The Next Step Forward in Multimodal LLM Alignment." *ICML 2025.*
>
> [2] Yu et al. "Every Question Has Its Own Value: Reinforcement Learning with Explicit Human Values." *arXiv preprint arXiv:2510.20187, 2025.*

---

> ### Author Response · Authors · 2025-11-21
> **Response to Reviewer 7qYS (3)**
>
> > Weakness 3: During data synthesis, different large models naturally produce different responses. However, the paper does not specify which models are used for the user agents, coordinator agent, or summarizer agent, nor the rationale behind these choices. Moreover, model differences in handling long contexts (e.g., varying user_number) can substantially affect the outputs generated by the coordinator and summarizer agents, potentially introducing additional variability into the synthesized data.
> >
> > Question 3: Which models are used for user agents, the coordinator, and the summarizer, and why? Do different context lengths (number of user numbers) affect the model selection for coordinator and summarizer agents?
>
> **3. Model Selection and Long Context Handling in Data Synthesis (Weakness 3 & Question 3)**
>
> We appreciate the request for clarification on our data synthesis pipeline, and we have supplemented the detailed architecture for data synthesis to the **Appendix A.3.1** (marked in blue).
>
> *   **Model Selection for Self-Distillation**: The core goal of our framework is to achieve **self-distillation** of collective social reasoning capabilities without external dependencies on stronger models. Therefore, as mentioned in Section 4.1, all MLLM agents in our framework, including the user agents, the coordinator agent, and the summarizer agent, **are implemented by the same model: Qwen2-VL-7B-Instruct**. This ensures that the collective reasoning is self-distilled into the base model.
>
> *   **Advanced Strategy for Long Context Handling**: Directly concatenating feedback from all users may have long-context challenges and affect the quality of the preference data synthesis. To address this, we design a **hierarchical summarization strategy** for the coordinator and summarizer agents:
>
> -  **First Level (Grouping by Profession)**: We first group the user responses (from the `potential correct` or `incorrect` set) into smaller sub-groups based on their professions. This design is because profession is an important factor influencing the way of thinking for social users.
>
> - **Second Level (Hierarchical Synthesis)**: The coordinator and summarizer agents then process each professional sub-group (typically 2-5 responses) to generate an initial summary. Finally, they synthesize these first-level summaries into a single, coherent, and comprehensive reasoning chain.
>
>   This hierarchical, profession-based approach allows the coordinator and summarizer agents to comprehensively incorporate the user attributes and effectively manage the long context of multi-perspective feedback, ensuring that diverse and unique insights are preserved and integrated.
>
> We hope these responses and new results have addressed your concerns. We are grateful for the opportunity to improve our paper based on your valuable feedback.

---

> ### Comment · Reviewer_7qYS · 2025-11-28
> **Thank you for the author's response.**
>
> I appreciate the authors' supplementary explanations regarding human evaluation, linear weighting, and the model, which greatly resolved my questions. I believe this is a highly complete paper, and I will maintain my score.

---

> > ### Author Response · Authors · 2025-11-28
> >
> > Thanks for your comments! We are glad that our answer addresses your concerns.

---

### Official Review · Reviewer_FGcP · 2025-10-31

**Soundness:** 3
**Presentation:** 3
**Contribution:** 3
**Rating:** 6
**Confidence:** 4

**Summary:**

This paper proposes a novel Multi-Social-Agent Self-Distillation framework for Multimodal Misinformation Detection (MMD), addressing the critical trade-off between the limited perspective of single-agent methods and the high computational cost of multi-agent systems. The core idea is to simulate a diverse society of MLLM agents to generate multi-perspective judgments on multimodal misinformation, synthesizing their collective feedback into high-quality Social Chain-of-Thought (SCoT) data. To effectively utilize this data, the authors introduce Social Correction Value-Driven Preference Optimization (SCPO), a new alignment algorithm. Extensive experiments on the MFC-Bench and MMFakeBench benchmarks demonstrate its effectiveness.

**Strengths:**

1 The central concept of "thinking as society" is highly innovative and well-motivated. The framework effectively addresses a genuine dilemma in the field by internalizing the benefits of multi-agent reasoning into a single, efficient model through self-distillation.

2 The paper introduces two key novel components: the generation of Social Chain-of-Thought (SCoT) data through simulated social feedback and the SCPO algorithm.

3 The experimental evaluation is thorough, covering two relevant benchmarks (MFC-Bench, MMFakeBench) and a wide range of baselines, including state-of-the-art open-source and proprietary models.

4 The paper is generally well-written and logically organized. The figures are effective in illustrating the core concepts and the data generation pipeline.

**Weaknesses:**

1 The quality of the final model is heavily dependent on the quality and diversity of the simulated social agents. The paper relies on profile-based prompting to create diverse agents, but the actual diversity and realism of the generated reasoning are not deeply analyzed or validated. There is a risk that the simulated "society" could exhibit unforeseen biases or lack true cognitive diversity, limiting the benefit of the approach.

2 The paper presents the full SCPO framework but lacks a crucial ablation study. It's unclear how much of the performance gain comes from the SCoT data itself versus the specific SCPO optimization algorithm.

3 The paper focuses on the impressive results but provides little analysis of where the proposed model still fails. Understanding the types of multimodal misinformation that remain challenging would provide deeper insight into the method's limitations and guide future work.

**Questions:**

1 Could the authors provide a more in-depth analysis or qualitative examples demonstrating the actual cognitive diversity of the reasoning generated by the different user agents? How can we be assured that the profile-based prompting doesn't lead to a homogenized or biased "society" that limits the effectiveness of the collective reasoning?

2 To better understand the individual contributions, could the authors include an ablation study comparing the performance when the high-quality SCoT data is trained using standard DPO or SFT, versus the proposed SCPO algorithm?

3 Could the authors provide an analysis of the failure cases on the benchmark datasets?

---

> ### Author Response · Authors · 2025-11-21
> **Response to Reviewer FGcP (1)**
>
> We sincerely appreciate your insightful comments and valuable suggestions, and we carefully address each of your comments and provide point-by-point responses below.
>
> > Weakness 1: The quality of the final model is heavily dependent on the quality and diversity of the simulated social agents. The paper relies on profile-based prompting to create diverse agents, but the actual diversity and realism of the generated reasoning are not deeply analyzed or validated. There is a risk that the simulated "society" could exhibit unforeseen biases or lack true cognitive diversity, limiting the benefit of the approach.
> >
> > Question 1: Could the authors provide a more in-depth analysis or qualitative examples demonstrating the actual cognitive diversity of the reasoning generated by the different user agents? How can we be assured that the profile-based prompting doesn't lead to a homogenized or biased "society" that limits the effectiveness of the collective reasoning?
>
> **1. Diversity and Realism of Simulated Social Agents (Weakness 1 & Question 1)**
>
> We thank the reviewer for raising this critical point about the quality of our social simulation. We address this concern from three perspectives: the foundation of our user profiles, a human validation study, and a quantitative analysis of cognitive diversity.
>
> *   **Foundation in Real-World Data**: As detailed in Appendix A.2.1, our `User Profile Database` is constructed using data from the OASIS platform, which itself is based on information from real-world social media users. This ensures that the demographic, professional, and interest-based attributes of our simulated agents are grounded in reality.
>
> *   **Human Validation of Reasoning Realism**: To directly validate the realism of the generated reasoning, **we immediately start a human evaluation study** as suggested. We randomly sample a diverse set of user responses across various topics (e.g., politics, news, environment, biology) and professional domains (e.g., law, service industry, IT, public administration), and ask human evaluators to rate the consistency between the reasoning style and the corresponding user profile on a 5-point Likert scale. **Our preliminary results are highly promising**:
>
>     | Rating         | 1 (Very Inconsistent) | 2     | 3     | 4          | 5 (Very Consistent) |
>     | :------------- | :-------------------- | :---- | :---- | :--------- | :------------------ |
>     | **Percentage** | 0.71%                 | 2.14% | 8.57% | **49.29%** | **39.29%**          |
>
>     These results indicate that **nearly 90% of the simulated responses were perceived by human judges as consistent (rating 4 or 5) with their assigned roles**. This provides strong evidence that our user agents generate realistic and diverse social perspectives. We will continue to gather more real investigation data for comprehensive evaluation of the actual diversity and realism of the generated reasoning.
>
> * **Quantitative Cognitive Diversity**: To quantitatively ensure cognitive diversity, we design a `user diversity` metric, as detailed in Appendix A.4.4.
>
>   $e_c = \frac{1}{U}\sum^{U}_{i=1}e_i$
>
>   $sim_c = \frac{1}{U}\sum^{U}_{i=1}\frac{e_c \cdot e_i}{|e_c| \cdot |e_i|}$
>
>   $dis = 1 - sim_c$
>
>   where $U$ denotes the user numbers for a sample, and $e_i$ denotes each persona embedding. This dispersion value represents the diversity of the sampled users. The stronger the user diversity, the broader the coverage of social simulation.
>
>   This analysis confirms that sampling 20 users per sample provides a strong balance between diverse perspectives and computational cost, preventing the simulation from becoming biased or uniform.
>
> Together, these three points provide a comprehensive validation of our social simulation's quality, diversity, and realism.

---

> ### Author Response · Authors · 2025-11-21
> **Response to Reviewer FGcP (2)**
>
> > Weakness 2: The paper presents the full SCPO framework but lacks a crucial ablation study. It's unclear how much of the performance gain comes from the SCoT data itself versus the specific SCPO optimization algorithm.
> >
> > Question 2: To better understand the individual contributions, could the authors include an ablation study comparing the performance when the high-quality SCoT data is trained using standard DPO or SFT, versus the proposed SCPO algorithm?
>
> **2. Contributions of SCoT Data and SCPO Algorithm (Weakness 2 & Question 2)**
>
> We appreciate the suggestion for validating the contributions of SCoT Data and SCPO Algorithm, which are the two core contributions of our work. We have conducted ablation studies to analyze their effects.
>
> * **Value of SCoT Data**: We supplement an ablation study to quantify the impact of our Social Chain-of-Thought (SCoT) data.
>
>   We train a baseline model (denoted as "Binary") using the same source data but only with the final binary "real/fake" labels, removing the detailed reasoning chains from the SCoT data. The performance is compared against our Supervised Fine-Tuning (SFT) model, which is trained on the positive examples from our SCoT data; and our SCPO model, which is trained on our entire SCoT preference data.
>
>   | Models | Size | Manipulation Acc (31K) | OOC Acc (2K) | Veracity Acc (2K) | Overall Acc (35K) |
>   | :----- | :--- | :--------------------- | :----------- | :---------------- | :---------------- |
>   | Binary | 7B   | 61.90                  | 74.35        | 81.15             | 63.71             |
>   | SFT    | 7B   | 63.14                  | 67.65        | 77.20             | **64.20**         |
>   | SCPO   | 7B   | 65.80                  | 74.60        | 80.75             | **67.15**         |
>
>   The results show that SCoT-based SFT and SCPO achieve +0.49% and +3.44% on the overall accuracy. Moreover, SFT and SCPO achieve +1.24% and +3.9% improvement on the Manipulation split, which is the most challenging and largest split (31K sample with 7 misinformation sub-types). This demonstrates that the detailed multi-perspective reasoning within SCoT data is particularly effective for MLLMs to handle complex and challenging types of misinformation.
>
>   Furthermore, **a model trained only on binary labels cannot generate a detailed reasoning process**. Therefore, the GPT-4 evaluation of reasoning quality on MFC-Bench is unavailable for this binary-trained model. This indicates that removing the CoT-style training data is not beneficial for the model interpretability. In contrast, the multi-perspective reasoning process generated by our SCPO model is a key aspect of our work's contribution, demonstrating the superiority of both overall performance and interpretability.
>
> * **Value of the SCPO Algorithm**: To isolate the contribution of our SCPO algorithm, **Table 3 in our manuscript provides the precise ablation study requested**. In this table, we compare the performance of standard SFT, SFT+DPO, and ORPO against our SCPO, **all trained on the exact same high-quality SCoT dataset**. The results clearly show that our SCPO model consistently outperforms these strong baselines. This confirms that our social correction value-driven mechanism construct a robust training signal based on diverse social feedbacks. Through this verifiable social signal, our model focuses its efforts on resolving the most complex cognitive gaps, providing more effective and stable optimization.

---

> ### Author Response · Authors · 2025-11-21
> **Response to Reviewer FGcP (3)**
>
> > Weakness 3: The paper focuses on the impressive results but provides little analysis of where the proposed model still fails. Understanding the types of multimodal misinformation that remain challenging would provide deeper insight into the method's limitations and guide future work.
> >
> > Question 3: Could the authors provide an analysis of the failure cases on the benchmark datasets?
>
> **3. Analysis of Failure Cases (Weakness 3 & Question 3)**
>
> We agree that analyzing failure cases is essential for understanding the model's limitations and guiding future research. We analyze several challenging cases, and supplement this detailed failure analysis to the **Appendix A.6** (marked in blue). Here are two representative examples:
>
> *   **Case 1: Overly Cautious Reasoning (Ground Truth: Real, Prediction: Fake)**
>     *   **Instance**: The text identifies "Trevor Birch" as an administrator for a football club. The image shows a man at a sports event.
>     *   **Analysis**: Our model correctly identifies the general scene (a man at a sports event) but concludes that the image and text do not align because there is no explicit visual evidence to confirm the man's identity as Trevor Birch.  In this case, the model adopts an **overly cautious stance**, failing to recognize the visual evidence and give the "Fake" conclusion. This shows a limitation for handling visual evidence which is contextually relevant but not completely explicit.
>
> *   **Case 2: Failure in Fine-Grained Entity Verification (Ground Truth: Fake, Prediction: Real)**
>     *   **Instance**: The text names four specific actors from the film "Last Vegas". The image shows four men who resemble the actors in a setting consistent with the film. The person on the far left is incomplete in the image, with only one arm.
>     *   **Analysis**: The model correctly identifies the general scene ("four men in a casual, beach-like setting") and assesses the overall visual-textual consistency as high. However, it **fails to perform the deeper, fine-grained verification** required to confirm that the individuals in the photo are indeed the claimed actors. The model's reasoning is dominated by the strong surface-level consistency, leading it to an incorrect "Real" conclusion. This reveals a challenge in moving beyond general scene understanding to specific, real-world entity verification.
>
> These failure cases suggest that future work could focus on improving the model's ability to balance cautious verification with plausible inference and enhancing its capacity for fine-grained, knowledge-based entity grounding.
>
> We hope these detailed responses, supported by new human evaluation data, existing ablations, and a thorough failure analysis, have fully addressed your concerns. We are grateful for the opportunity to improve our work based on your valuable feedback.

---

> ### Comment · Reviewer_FGcP · 2025-11-26
> **I will keep my score**
>
> The author has addressed my questions satisfactorily.

---

> > ### Author Response · Authors · 2025-11-26
> >
> > Thanks for your comments! We are glad that our answer addresses your concerns.

---

### Official Review · Reviewer_2dcz · 2025-11-01

**Soundness:** 3
**Presentation:** 3
**Contribution:** 3
**Rating:** 6
**Confidence:** 2

**Summary:**

The paper proposes a novel framework called Multi-Social-Agent Self-Distillation (MSA-SD) to enhance the reasoning capability and social robustness of Multimodal Misinformation Detection (MMD). Based on the Qwen2-VL-7B model, the authors simulate feedback from multiple social roles to generate Social Chain-of-Thought (SCoT) data and introduce a new preference alignment algorithm, Social Correction Value-Driven Preference Optimization (SCPO), which dynamically focuses on samples with large social cognition divergences. Experiments conducted on two multimodal fact-checking benchmarks, MFC-Bench and MMFakeBench, demonstrate that the proposed framework significantly outperforms multi-agent methods as well as various open-source and closed-source MLLM models.

**Strengths:**

1.Innovative framework design. The paper proposes a multi-social-agent self-distillation mechanism that integrates multi-perspective social reasoning into a single model, demonstrating strong novelty.

2.Introduction of the social correction value sc(x). The dynamic weighting of samples through sc(x) provides a new, verifiable optimization signal. The “thinking as society” paradigm holds potential for broad conceptual influence.

3.Comprehensive and significant experimental validation. Based on Qwen2-VL, SCPO achieves a +9.9% accuracy improvement, significantly surpassing InternVL3 and Qwen2.5-VL. Using GPT-4 to evaluate reasoning quality, SCPO achieves the best performance across four dimensions—misleadingness, informativeness, logicality, and readability.

**Weaknesses:**

1.Insufficient ablation and error analysis. Although multiple optimization methods are compared, the paper does not quantify how the quality of SCoT data affects performance, nor analyze the independent contributions of each module (e.g., coordinator/summarizer). In Sec. 4.2 (Table 1), results are reported without standard deviations or significance tests. The paper also fails to present performance differences across samples with varying difficulty levels (high/low sc(x)).

2.Missing experimental details. The concept of “thinking as society” lacks a concrete definition or illustrative examples. Hardware environment, training duration, and batch size should be explicitly reported. Key results in tables should include ± standard deviation or t-test significance. The differences between open and closed prompts should be analyzed, along with a discussion of model generalization.

**Questions:**

1.Insufficient ablation and error analysis. Although multiple optimization methods are compared, the paper does not quantify how the quality of SCoT data affects performance, nor analyze the independent contributions of each module (e.g., coordinator/summarizer). In Sec. 4.2 (Table 1), results are reported without standard deviations or significance tests. The paper also fails to present performance differences across samples with varying difficulty levels (high/low sc(x)).

2.Missing experimental details. The concept of “thinking as society” lacks a concrete definition or illustrative examples. Hardware environment, training duration, and batch size should be explicitly reported. Key results in tables should include ± standard deviation or t-test significance. The differences between open and closed prompts should be analyzed, along with a discussion of model generalization.

---

> ### Author Response · Authors · 2025-11-21
> **Response to Reviewer 2dcz (1)**
>
> We sincerely appreciate your insightful comments and valuable suggestions, and we carefully address each of your comments and provide point-by-point responses below.
>
> > Weakness 1 and Question 1: Insufficient ablation and error analysis. Although multiple optimization methods are compared, the paper does not quantify how the quality of SCoT data affects performance, nor analyze the independent contributions of each module (e.g., coordinator/summarizer). In Sec. 4.2 (Table 1), results are reported without standard deviations or significance tests. The paper also fails to present performance differences across samples with varying difficulty levels (high/low sc(x)).
>
> **1. Ablation and Error Analysis**
>
> We appreciate the detailed suggestions for improving our experimental analysis. We have conducted new experiments and analysis to address each point.
>
> * **Quality of SCoT Data:**
>
>   Following the valuable suggestion, we supplement an ablation study to quantify the impact of our Social Chain-of-Thought (SCoT) data. We train a baseline model (denoted as "Binary") using the same source data but only with the final binary "real/fake" labels, removing the detailed reasoning chains from the SCoT data. The performance is compared against our Supervised Fine-Tuning (SFT) model, which is trained on the positive examples from our SCoT data; and our SCPO model, which is trained on our entire SCoT preference data.
>
>   | Models | Size | Manipulation Acc (31K) | OOC Acc (2K) | Veracity Acc (2K) | Overall Acc (35K) |
>   | :----- | :--- | :--------------------- | :----------- | :---------------- | :---------------- |
>   | Binary | 7B   | 61.90                  | 74.35        | 81.15             | 63.71             |
>   | SFT    | 7B   | 63.14                  | 67.65        | 77.20             | **64.20**         |
>   | SCPO   | 7B   | 65.80                  | 74.60        | 80.75             | **67.15**         |
>
>   The results show that SCoT-based SFT and SCPO achieve +0.49% and +3.44% on the overall accuracy. Moreover, SFT and SCPO achieve +1.24% and +3.9% improvement on the Manipulation split, which is the most challenging and largest split (31K sample with 7 misinformation sub-types). This demonstrates that the detailed multi-perspective reasoning within SCoT data is particularly effective for MLLMs to handle complex and challenging types of misinformation.
>
>   Furthermore, **a model trained only on binary labels cannot generate a detailed reasoning process**. Therefore, the GPT-4 evaluation of reasoning quality on MFC-Bench is unavailable for this Binary model. This indicates that removing the CoT-style training data is not beneficial for the model interpretability. In contrast, the multi-perspective reasoning process generated by our SCPO model is a key aspect of our work's contribution, demonstrating the superiority of both overall performance and interpretability.
>
> * **Independent Contributions of Modules (Coordinator/Summarizer):**
>
>   We appreciate the suggestion to analyze the independent contributions of the coordinator and summarizer agents.  We would like to clarify that these agents are **functionally integral components of our data synthesis pipeline**.  Their roles are essential for transforming the unstructured, multi-perspective feedback from user agents into the high-quality preference pairs (`chosen` vs.  `rejected` SCoTs) required for preference optimization algorithms like DPO, ORPO, and our SCPO.
>
>   * Without the **Coordinator Agent**, we cannot synthesize the coherent, positive reasoning process (the `chosen` samples).
>   * Without the **Summarizer Agent**, we cannot construct the representative flawed reasoning process (the `rejected` samples).
>
>   Therefore, removing either module would prevent the creation of the SCoT preference dataset itself, indicating that a direct ablation for these two important agents may be infeasible. They are necessary components for the entire learning framework, which is designed to distill the collective reasoning into a unified model.

---

> ### Author Response · Authors · 2025-11-21
> **Response to Reviewer 2dcz (2)**
>
> * **Significance Tests and Standard Deviations:**
>
>   We appreciate the suggestion to analyze the result stability. We would like to first clarify that our main experiments used a fixed random seed (42) for reproducibility. Moreover, we have now conducted **three separate inference runs with our final SCPO model** to assess its stability.
>
>   | Run           | Size | Manipulation Acc (31K) | OOC Acc (2K) | Veracity Acc (2K) | Overall Acc (35K) |
>   | :------------ | :--- | :--------------------- | :----------- | :---------------- | :---------------- |
>   | **Run 1**     | 7B   | 65.80                  | 74.60        | 80.75             | 67.15             |
>   | **Run 2**     | 7B   | 65.62                  | 74.60        | 81.55             | 67.05             |
>   | **Run 3**     | 7B   | 65.67                  | 75.55        | 81.25             | 67.13             |
>   | **Std. Dev.** |      | 0.09                   | 0.55         | 0.41              | 0.05              |
>
>   The results show very low variance, with the overall accuracy remaining stable around **67.1%**. This demonstrates the robustness of our model's predictions.
>
> * **Performance with Varying Difficulty Levels (high/low `sc(x)`):**
>
>   We appreciate this insightful question. We would like to first clarify that `sc(x)` is a **training-time signal** derived from social simulation. At the inference stage, since we have distilled multi-perspective reasoning abilities into a unified model, we directly use the SCPO model for evaluation.
>
>   While `sc(x)` is a training-time signal, we can examine the different MFC-Bench splits to analyze the performance on test samples of varying difficulty. The **Manipulation** split is widely considered the most challenging task in MFC-Bench. It contains the vast majority of the data (31K samples) and covers 7 diverse and subtle manipulation types (e.g., face swap, photoshop, style transfer).  In contrast, `OOC` and `Veracity` focus on more direct and relatively simple forms of inconsistency.
>
>   As shown in Table 1 of our paper, our SCPO framework demonstrates consistent performance gains across all splits, including the most difficult Manipulation task. This indicates that our method effectively enhances the model's ability to handle challenging samples by internalizing social reasoning.

---

> ### Author Response · Authors · 2025-11-21
> **Response to Reviewer 2dcz (3)**
>
> > Weakness 2 and Question 2: Missing experimental details. The concept of “thinking as society” lacks a concrete definition or illustrative examples. Hardware environment, training duration, and batch size should be explicitly reported. Key results in tables should include ± standard deviation or t-test significance. The differences between open and closed prompts should be analyzed, along with a discussion of model generalization.
>
> **2. Missing Experimental Details**
>
> We appreciate the reviewer's request for more detailed explanations. We will clarify these points in the revised manuscript.
>
> *   **Definition of "Thinking as Society":** We provide a more concrete definition of "thinking as society" as an advanced reasoning paradigm designed to enable MLLMs to simulate human social intelligence, allowing a single model to reason from the perspectives of multiple social roles, thereby realizing a deep understanding and decision-making for MMD task. We have supplemented this definition in the **Introduction** (marked in blue).
>
> *   **Hardware and Training Details:** Key experimental details are crucial. We have updated the comprehensive hardware and training details in **Appendix A.3.2** (marked in blue), including the hardware used (2 NVIDIA A800 80GB GPUs), training parameters (batch size of 4 with 4 gradient accumulation steps), and total training duration (approx. 8 hours).
>
> *   **Open/Closed Prompts and Generalization:** We provide the differences between open and closed prompting in **Section 4.1** and **Appendix A.2.3**. Open prompting evaluates generalization by using a single, universal prompt for all misinformation types, simulating real-world scenarios where the forgery type is unknown. Closed prompting evaluates in-domain performance by providing task-specific instructions in the system prompts. To explicitly demonstrate our model's generalization capabilities, all results for our SCPO model are reported under the more challenging **open-prompting setting**.
>
> We hope these responses and new results have addressed your concerns. We are grateful for the opportunity to improve our paper based on your valuable feedback.

---

### Meta-Review · Area_Chair_BsXg · 2026-01-07

**Summary:**

This paper proposes a multi-social-agent self-distillation framework for multimodal misinformation detection, combining SCoT data synthesis with a preference optimization method, SCPO. Reviewers agree that the problem is timely and relevant, and that the overall approach is technically coherent. However, concerns were raised regarding the conceptual distinctiveness of the multi-social-agent formulation, the marginal performance gains of SCPO relative to existing preference optimization methods, and the strength of evidence supporting claims of social realism and diversity. The rebuttal improves experimental clarity and provides additional analyses, moving the paper closer to the acceptance boundary without fully resolving all conceptual concerns.

**Reviewer Concerns:**

**Concerns Addressed:**
* The authors clarified training details, hardware settings, prompting strategies, and data curation, resolving earlier reproducibility and reporting gaps.
* New experiments disentangle the effects of binary supervision, SCoT-based supervised fine-tuning, and SCPO, supporting the claim that richer supervision is the dominant source of improvement.
* Added qualitative failure cases clarify remaining limitations, particularly in fine-grained entity verification and overly cautious reasoning.
* Human role-consistency evaluation and quantitative dispersion metrics provide partial support that simulated agents are not trivially degenerate, though validation remains indirect.

**Concerns Still Outstanding:**
* The multi-social-agent paradigm remains closer to structured multi-persona sampling and hierarchical synthesis than to interactive multi-agent reasoning, limiting the strength of the conceptual leap.
* Improvements over ORPO are consistent but modest, reinforcing reviewer concerns that performance gains are driven primarily by SCoT data quality rather than the specific weighting formulation.
* While role consistency is validated, there is still limited direct comparison to real human judgments in misinformation assessment, leaving the societal grounding of the approach only partially substantiated.

**Reviewer Scores:**

Reviewer 2dcz: 6 -> 6
* Empirical rigor improved, but core conceptual reservations persist.

Reviewer FGcP: 6 -> 6
* Technical questions were addressed, yet concerns about agent realism and attribution of gains remain.

Reviewer 7qYS: 6 -> 6
* Clarifications strengthen justification, though SCPO’s incremental impact limits score change.

Reviewer NiZd: 4 -> 6
* Additional transparency and analysis alleviate novelty and methodology concerns.

**Overall:**
The rebuttal substantially improves empirical clarity, ablation coverage, and methodological transparency, and the approach is technically sound and carefully evaluated. While conceptual positioning and the incremental nature of the SCPO contribution remain debated, these concerns primarily affect framing and degree of novelty rather than correctness. The paper sits near the acceptance boundary, but the strengthened experimental grounding and demonstrated effectiveness justify acceptance. I therefore recommend accept.

---

### Decision · Program_Chairs · 2026-01-26

Accept (Poster)